# The manifold structure of limb coordination in walking *Drosophila*

Brian D DeAngelis[1†], Jacob A Zavatone-Veth[2†], Damon A Clark[1,2,3,4]*

[1]Interdepartmental Neuroscience Program, Yale University, New Haven, United States; [2]Department of Physics, Yale University, New Haven, United States; [3]Department of Molecular, Cellular, and Developmental Biology, Yale University, New Haven, United States; [4]Department of Neuroscience, Yale University, New Haven, United States

**Abstract** Terrestrial locomotion requires animals to coordinate their limb movements to efficiently traverse their environment. While previous studies in hexapods have reported that limb coordination patterns can vary substantially, the structure of this variability is not yet well understood. Here, we characterized the symmetric and asymmetric components of variation in walking kinematics in the genetic model organism *Drosophila*. We found that *Drosophila* use a single continuum of coordination patterns without evidence for preferred configurations. Spontaneous symmetric variability was associated with modulation of a single control parameter—stance duration—while asymmetric variability consisted of small, limb-specific modulations along multiple dimensions of the underlying symmetric pattern. Commands that modulated walking speed, originating from artificial neural activation or from the visual system, evoked modulations consistent with spontaneous behavior. Our findings suggest that *Drosophila* employ a low-dimensional control architecture, which provides a framework for understanding the neural circuits that regulate hexapod legged locomotion.

**\*For correspondence:**
damon.clark@yale.edu

[†]These authors contributed equally to this work

**Competing interests:** The authors declare that no competing interests exist.

## Introduction

Legged locomotion requires flexible coordination of limbs in varied, changing environments. This coordination has been studied in a variety of model organisms, from camels and rhinoceros to ferrets, rats, beetles, and ants (*Alexander and Jayes, 1983*; *Büschges et al., 2008*). In hexapods, many studies have noted considerable variability in limb movements (*Ayali et al., 2015*; *Mendes et al., 2013*; *Pereira et al., 2019*; *Strauß and Heisenberg, 1990*; *Wosnitza et al., 2013*; *Zollikofer, 1994*), but the structure and origin of this variability remain unclear. A comprehensive characterization of this variability is required to understand locomotor control in insects (*Krakauer et al., 2017*). To understand how insects coordinate their limbs during terrestrial locomotion, we investigated the variability in walking patterns in the fruit fly *Drosophila*, where we could record behavior in many flies and manipulate neural signals with genetic tools.

In many animals, variability in walking patterns takes the form of distinct locomotor gaits (*Alexander, 1989*; *Alexander and Jayes, 1980*; *Alexander and Jayes, 1983*; *Srinivasan and Ruina, 2006*; *Thorstensson and Roberthson, 1987*). In the study of hexapod locomotion, it is unclear how the observed variability in walking coordination relates to potential preferred gaits or to forms of continuous variability. Although some studies report distinct gaits in insects (*Ayali et al., 2015*; *Bender et al., 2011*; *Burrows, 1996*; *Graham, 1972*; *Mendes et al., 2013*; *Pereira et al., 2019*), many note a high degree of variability in limb coordination, with intermediate patterns intermixed with movements that resemble canonical gaits (*Ayali et al., 2015*; *Mendes et al., 2013*; *Pereira et al., 2019*; *Strauß and Heisenberg, 1990*; *Szczecinski et al., 2018*; *Wosnitza et al., 2013*; *Zollikofer, 1994*). *Drosophila* have been reported to use different coordination patterns at

low and high walking speeds, with significant variability around these canonical patterns (*Mendes et al., 2013*; *Pereira et al., 2019*; *Strauß and Heisenberg, 1990*; *Szczecinski et al., 2018*; *Wosnitza et al., 2013*). The nature of this variability has not been fully investigated, as these studies focused primarily on small numbers of individual bouts of walking.

When walking speed is varied, it represents a form of *symmetric* variability in walking patterns. These walking patterns are themselves often antisymmetric, with legs moving out of phase, but we refer to modulations of these patterns as symmetric if they are equal on the two sides of the body. This symmetry maintains heading. Locomotion also contains *asymmetric* variability, in which modulations are unequal on the two sides of the body. Such asymmetry is required to change heading. In hexapods, measurements of limb kinematics (*Cruse et al., 2009*; *Domenici et al., 1998*; *Dürr and Ebeling, 2005*; *Franklin et al., 1981*; *Frantsevich and Mokrushov, 1980*; *Graham, 1972*; *Gruhn et al., 2009*; *Mu and Ritzmann, 2005*; *Strauß and Heisenberg, 1990*; *Zollikofer, 1994*; *Zolotov et al., 1975*) and dynamics (*Jindrich and Full, 1999*) have described small, limb-specific modulations of movement associated with turning. As in the study of symmetric variability, these studies focused primarily on individual bouts of locomotion, and most employed tethered preparations. To understand how limb movements are modulated during turns, and how symmetric and asymmetric forms of variability are related to one another, it is necessary to observe full distributions of behavior in intact and unrestrained animals.

In this study, we generated large datasets of locomotor behavior using a simple method for robustly tracking the body and limb movements of freely-walking *Drosophila*. By analyzing hundreds of thousands of steps, we found that the locomotor variability in flies exists on a single continuous manifold, without evidence for discrete preferred patterns. Symmetric variability could be described by changes to a single global parameter governing walking speed. In contrast, turning involves small, asymmetric, and limb-specific modulations that are precisely timed relative to the underlying symmetric pattern. When slowing was evoked by optogenetic activation of command neurons or by visual stimuli, the resulting symmetric modulations of limb movement were consistent with the manifold present in spontaneous walking. The structure of the variability in spontaneous and evoked coordination patterns suggests a simple model for limb coupling during locomotion.

## Results

### A simple automated method robustly identifies limb positions

We constructed a planar arena in which freely-walking flies were illuminated from above and could be filmed from below in silhouette by a high-speed camera at 150 frames per second (*Figure 1A*). Within each frame, flies were first located and then rotated into a common orientation (see Materials and methods). From the centroid positions and heading angles, we computed the fly's forward velocity (parallel to its heading), its lateral velocity (perpendicular to its heading), and its yaw velocity, which we denote as $v_{\parallel}$, $v_{\perp}$, and $v_r$, respectively (*Figure 1B*). Exploiting the stereotyped anatomy of flies, we then used a simple linear model to identify the position of each limb in each frame (see Materials and methods, *Figure 1—figure supplement 1*, *Table 1*, and *Videos 1–3*). To ensure that correlations between limb poses in our training set did not systematically bias predicted limb positions, we required the model to predict limb positions from image regions limited to each limb's approximate range of motion (*Figure 1—figure supplement 1*). We trained our model on a set of 5000 hand-annotated images, and used 10-fold cross-validation to test for overfitting. This cross-validation showed that the mean error between predicted and hand-annotated limb positions was 0.15 mm (3.5 pixels) (*Figure 1—figure supplement 1*). Importantly, the accuracy of this simple linear model in our experimental context was comparable to that achieved by published deep neural network methods (*Mathis et al., 2018*; *Pereira et al., 2019*) (*Figure 1—figure supplement 1*).

### Measurements of spontaneous wild-type walking are consistent with previous studies

Many previous studies have investigated the statistics of walking in *Drosophila* (*Berman et al., 2014*; *Bidaye et al., 2014*; *Branson et al., 2009*; *Geurten et al., 2014*; *Kain et al., 2013*; *Katsov and Clandinin, 2008*; *Katsov et al., 2017*; *Martin, 2004*; *Mendes et al., 2013*; *Strauß and Heisenberg, 1990*; *Wosnitza et al., 2013*). As the flies walked in the arena, their heading, lateral, and forward

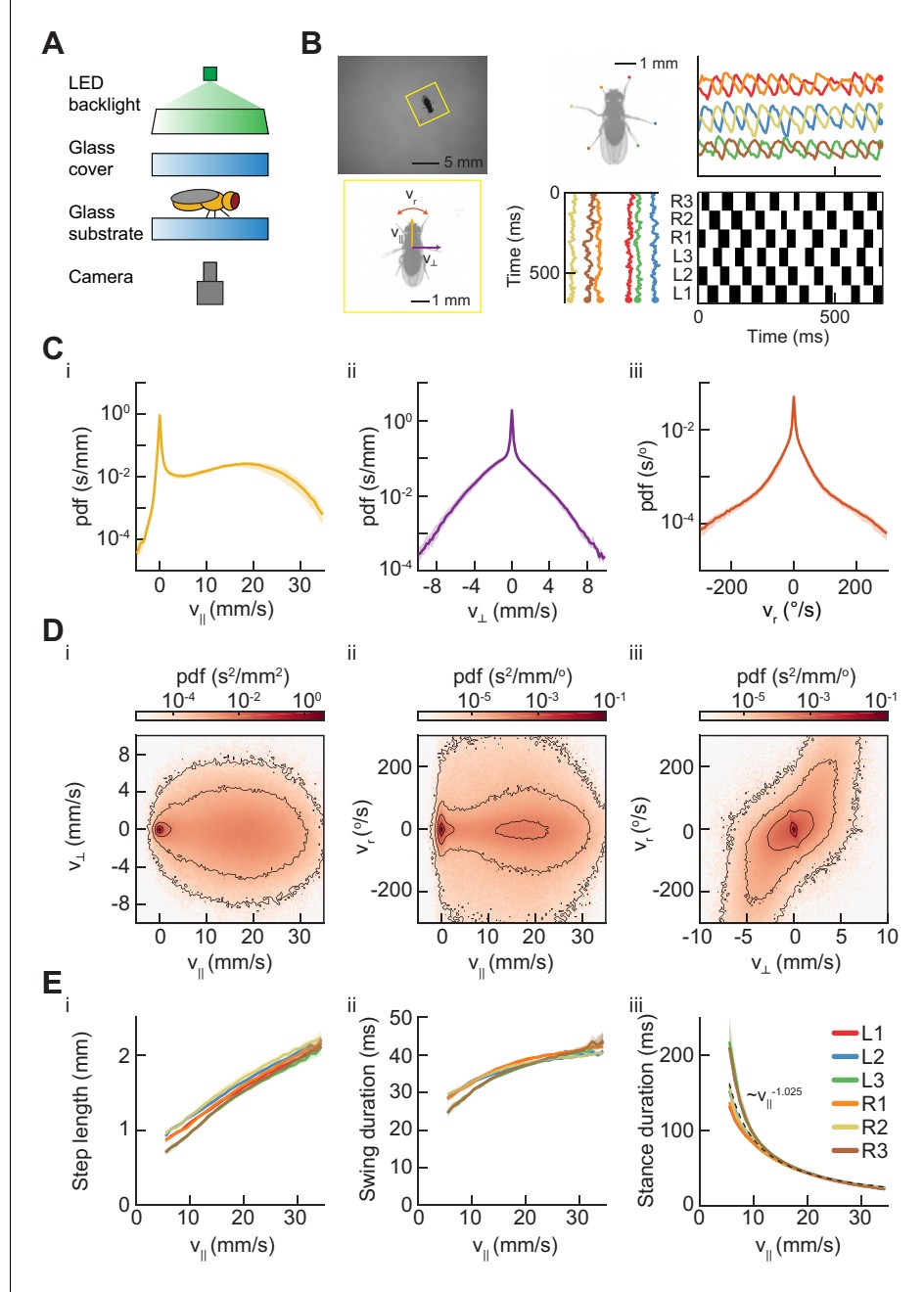

**Figure 1.** Measurements of body and limb kinematics in freely-walking *Drosophila*. (**A**) Schematic of experimental setup. Fruit flies walk in a circular arena while illuminated from above and tracked from below using a high-speed camera (150 fps). (**B**) Data transformations from unstructured video to time series variables. Flies are identified in raw camera frames. Individual fly-frames are then grouped and aligned across sequential camera images. Limb positions relative to the fly's center of mass are converted into time series variables describing limb movements in the egocentric fly frame. The fly's limbs are binarized into individual periods of swing (*black*) and stance (*white*). (**C**) Probability density functions (PDFs) of three components of body movement: forward walking velocity ($v_\parallel$), lateral walking velocity ($v_\perp$), and rotational velocity ($v_r$). (**D**) Joint distributions of body velocity components: forward velocity vs. lateral velocity, forward velocity vs. rotational velocity, and lateral velocity vs. rotational velocity. (**E**) Mean stepping statistics as a function of forward velocity. Step length in the camera frame increases linearly with forward velocity. Swing duration is roughly constant as forward velocity increases when compared to changes in stance duration. Stance duration decreases inversely with increasing forward velocity.

*Figure 1 continued on next page*

**Table 1.** Fly strains.

|  | Genotype | Source | Experiment | Figure |
|---|---|---|---|---|
| Wild type | +; +; + | *Gohl et al., 2011* | Free-walking; Visual stimulus induced slowing | 1–8 |
| Moonwalker > Chrimson | +; +; VT-050660-Gal4/UAS-Chrimson | *Bidaye et al., 2014* | Optogenetic induced slowing | 8 |

velocities oscillated continuously due to the periodicity of limb forces (*Figure 1—figure supplement 2*). Since we were interested in longer-timescale modulations of the path, we smoothed the centroid kinematics to eliminate this oscillation (*Figure 1—figure supplement 2*, see Materials and methods) (*Katsov and Clandinin, 2008*; *Katsov et al., 2017*). Consistent with prior studies, we observe forward velocities between −1.3 and 30.4 mm/s (2.5th and 97.5th percentiles), with peaks in the distribution of forward walking speeds at 0 mm/s and ~17.5 mm/s (*Figure 1Ci*). Variation across studies in the measured distributions of walking speeds may be due to variation in experimental conditions that affect the motivational state of the fly, such as temperature and hunger (*Chadha and Cook, 2014*; *Crill et al., 1996*; *Soto-Padilla et al., 2018*).

Our large dataset enabled us to characterize the full distributions of centroid kinematics, rather than measuring only a small number of individual examples. The distributions of lateral and angular velocities were roughly symmetric, illustrating that we do not observe population-level handedness in *Drosophila* walking (*Figure 1C*). The joint distribution of the forward velocity $v_\parallel$ and the yaw velocity $v_r$ showed that flies turn at a broad range of yaw rates across many forward speeds (*Figure 1Di–ii*). Crab-walking, in which the fly's heading vector is no longer tangent to its path, predominately occurs during high-yaw-rate turns at slow to moderate forward velocities (*Figure 1Di–iii*). These characteristics of aggregate kinematic behavior are consistent with those reported in previous studies (*Strauß and Heisenberg, 1990*; *Geurten et al., 2014*; *Katsov and Clandinin, 2008*; *Katsov et al., 2017*).

Our measurements of limb kinematic parameters were also broadly consistent with previous studies. Importantly, we restricted this and subsequent analyses of limb coordination to locomotion by excluding flies moving below 0.5 mm/s. This criterion excluded flies that were stopped or grooming, behaviors that involve interesting patterns of limb coordination (*Seeds et al., 2014*), but lie outside of the focus of this study. In our data, step length in the stationary frame of the camera increased roughly linearly with forward walking speed (*Mendes et al., 2013*) (*Figure 1Ei*). Across walking speeds, swing duration increased, but this modulation is small in comparison to the change in stance duration (*Figure 1Eii–iii*). Stance duration was approximately inversely proportional to forward walking speed ($\tau_{\text{stance}} \sim v_\parallel^{-1.025}$, $R^2 = 0.59$) (see Materials and methods) (*Figure 1Eiii*). Therefore, our data were broadly consistent with previous observations that walking speed differences are dominated by changes in stance duration, while swing duration remains relatively constant (*Wilson, 1966*; *Wosnitza et al., 2013*; *Dürr et al., 2018*; *Mendes et al., 2013*).

## Swing-stance patterns change in a speed-dependent manner

In hexapod locomotion, studies have described three canonical gaits (*Figure 2A–B*) (*Bender et al., 2011*; *Burrows, 1996*; *Collins and Stewart, 1993*; *Graham, 1972*; *Spirito and Mushrush, 1979*; *Wilson, 1966*). In tripod gait, two groups of three limbs swing simultaneously, creating two alternating configurations of swing and stance. In tetrapod gait, three groups of two limbs swing in sequence,

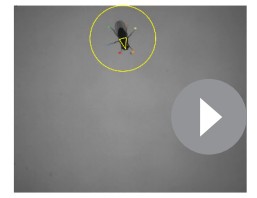
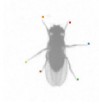

**Video 1.** This movie shows a fly walking in our arena with annotated body and limb features. Video shows annotations in both the camera and egocentric frame of the fly. Body orientation is indicated with a *yellow* triangle. Limb positions are *red* (L1), *blue* (L2), *green* (L3), *orange* (R1), *yellow* (R2), and *brown* (R3). https://elifesciences.org/articles/46409#video1

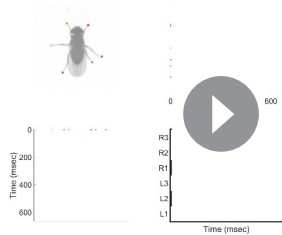

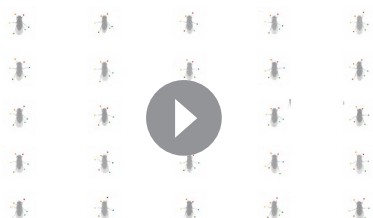

**Video 2.** This movie shows a fly in its egocentric frame with annotations for each of the individual limbs. Limb position variables are shown as time series in the direction parallel and perpendicular to the fly's major axis. Limb positions are *red* (L1), *blue* (L2), *green* (L3), *orange* (R1), *yellow* (R2), and *brown* (R3). Swing (*black*) and stance (*white*) events for each of the limbs are shown as a function of time.
https://elifesciences.org/articles/46409#video2

**Video 3.** This movie shows a grid of 25 fly trajectories seen in the egocentric frame. Limb positions are annotated with *red* (L1), *blue* (L2), *green* (L3), *orange* (R1), *yellow* (R2), and *brown* (R3).
https://elifesciences.org/articles/46409#video3

while in wave gait, each limb swings individually. In general, distinct gaits can be distinguished by discontinuous changes in one or more parameters of the coordination pattern (*Alexander, 1989*; *Srinivasan and Ruina, 2006*). However, defining gaits in terms of discontinuous transitions can be problematic in the presence of measurement noise or behavioral variability, which can blur sharp transitions. Instead, one may characterize the distribution of walking coordination patterns, where distinct gaits correspond to multiple modes in the distribution (*Hoyt and Taylor, 1981*).

To examine the structure of limb coordination in flies, we first investigated the structure of the swing-stance patterns used at different forward walking speeds (*Figure 2C*). We determined whether a limb was in swing or stance phase by measuring its frame-to-frame displacement in the camera frame. If a limb position moved more than roughly our error in positional measurement (~0.13 mm in the camera frame) then we considered the limb to be in swing phase (see Materials and methods). Movements below this threshold were classified as stance phase. Thus, for each fly-frame, we generated a six-digit binary vector describing the swing-stance configuration of the fly's six limbs.

These individual configurations can be categorized by how many limbs are in stance phase. Previous studies have reported an enrichment of the three-foot stance category at faster forward walking speeds and a corresponding enrichment of the four-foot stance category at slower forward walking speeds (*Mendes et al., 2013*; *Pereira et al., 2019*; *Strauß and Heisenberg, 1990*; *Wosnitza et al., 2013*). These studies suggested that the proportion of feet in stance at different speeds were consistent with flies using tetrapod gaits at lower speeds and a tripod gait at higher speeds. Our data also showed speed-dependent enrichments of five- and four-foot stance categories at slow speeds, and enrichments of the three-foot stance category for flies walking at faster speeds (*Figure 2C*).

To understand the structure of these different stance categories, we investigated the specific configurations of limbs in each of the three-, four-, and five-foot stance categories. For each of the canonical gaits, one would expect the constituent discrete limb configurations to occur in roughly equal proportions (*Figure 2B*). In the case of the three-foot stance category, the two configurations that comprise the canonical tripod gait occur in roughly equal proportions at all forward walking speeds (*Figure 2D*). In contrast, different configurations of the five-foot and four-foot stance categories do not occur with equal abundance across walking speeds (*Figure 2D*). Interestingly, the four-foot stance configurations that occur with greatest frequency are those that are a single swing-stance transition away from a canonical tripod gait.

## A large fraction of four- and five-foot stance configurations are transient

To investigate the dynamics of stance configurations in our data, we examined the dwell times in all individual stance configurations within each canonical gait (*Figure 2E*). We defined the dwell time as the number of contiguous frames in the configuration. If each canonical gait existed in the data, one

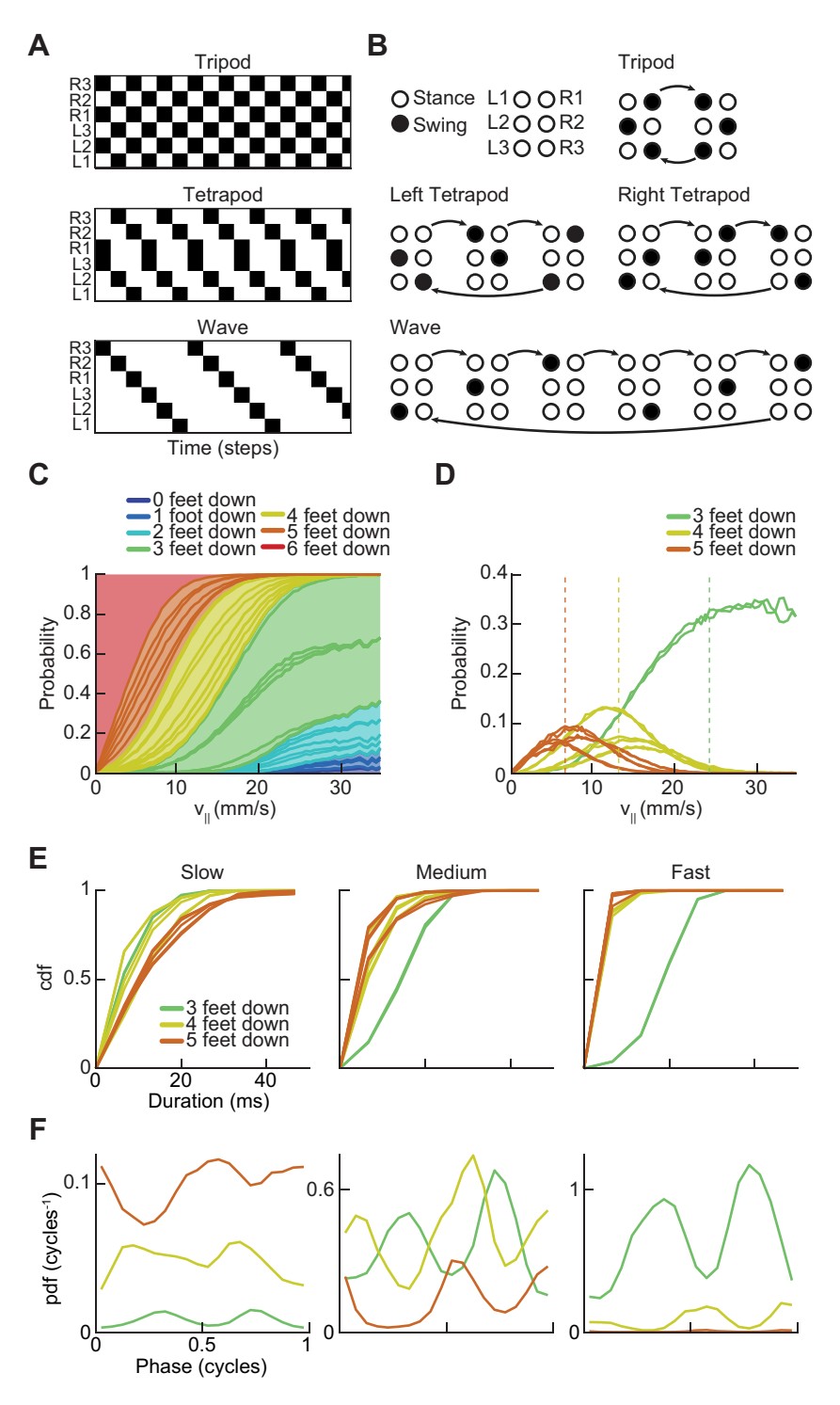

**Figure 2.** *Drosophila* use a two-cycle limb coordination pattern across all walking speeds. (**A**) Canonical hexapod gaits. In all gaits, limb swings (*black*) propagate ipsilaterally posterior to anterior on each side of the fly. Tripod gait is defined by three limbs simultaneously swinging at each point in time. Tetrapod gait is defined by two limbs simultaneously swinging at each point in time. Wave gait is defined by each limb swinging individually. (**B**) Each canonical gait contains a different number of distinct stance configurations, and correspondingly a different number of transitions between these configurations. Tripod gait contains two distinct configurations, tetrapod gait contains three distinct configurations, and wave gait contains six distinct configurations. The ordering of configurations within a cycle is defined by the posterior-to-anterior propagation of ipsilateral swing events (*black*).

*Figure 2 continued on next page*

*Figure 2 continued*

(C) Probability of number of feet in stance at each forward walking velocity. Segments within each color represent different configurations within each category. Number of feet in stance decreases as forward walking velocity increases. (D) Probability of canonical stance configurations (from (B)) as a function of forward velocity: tripod (*green*), tetrapod (*yellow*), wave gait (*orange*). Dashed lines indicate the forward speed with maximum probability of each stance category: 24 mm/s, 13 mm/s, and 7 mm/s 3-foot-down, 4-foot-down, and 5-foot-down categories, respectively. (E) Cumulative distribution function (CDF) of stance durations for all canonical gait configurations: tripod (*green*), tetrapod (*yellow*), and wave gait (*orange*). Configuration durations are visualized for bottom, middle, and top thirds of forward velocity distribution: slow walking (0–10.2 mm/s), medium walking (10.2–19 mm/s), and fast walking (>19 mm/s). Tetrapod and wave gait configurations are predominantly transient in all but the slowest walking condition. (F) Relative probability of number of feet in stance as a function of midlimb phase grouped by walking velocity: slow (0–10.2 mm/s), medium (10.2–19 mm/s) and fast walking (>19 mm/s). These probability density functions are normalized such that the integral over phase of the sum of the distributions conditioned on the number of feet down is equal to unity. The number of feet in stance varies as a function of limb phase with a periodicity of two per limb cycle.

The online version of this article includes the following figure supplement(s) for figure 2:

**Figure supplement 1.** Estimating limb phases.
**Figure supplement 2.** A two-cycle coordination pattern is used across all walking speeds.

would expect that the dwell time in each stance configuration associated with the gait would approximately match the average swing duration of the limbs. Because previous studies reported that *Drosophila* preferentially use tetrapod and tripod gaits at slow and fast speeds respectively, we partitioned the data into thirds based on forward walking speed (*Mendes et al., 2013*; *Pereira et al., 2019*; *Wosnitza et al., 2013*). At slow speeds ($v_{\parallel} \leq 10.2$ mm/s), while many instances of canonical configurations were transient (one frame or ~6.7 ms), there existed longer durations for all tripod, tetrapod, and wave gait configurations (*Figure 2E*). At higher speeds, virtually all tetrapod and wave gait configurations were transient, lasting only one frame.

Next, we found the most frequent forward velocities within each stance category, which were 7 mm/s, 13 mm/s, and 24 mm/s for 5-foot, 4-foot, and 3-foot stance categories, respectively (*Figure 2D*). If each canonical configuration represented a distinct, preferred gait, then we might expect these walking speeds to occur with greater frequency than nearby walking speeds, as is observed in horses (*Hoyt and Taylor, 1981*). Yet, no peaks in the distribution of forward velocities exist in these locations, suggesting that this phenomenon is not present in flies (*Figure 1Ci*).

## The abundance of each stance category peaks twice per single-limb stride

Each canonical gait contains a different number of discrete limb configurations and thus a different number of transitions between these configurations (*Figure 2B*). In canonical gaits, swinging limbs move simultaneously, but deviations in step timing can transiently change the number of feet down. For tripod gait, changes in the number of feet down would occur twice per cycle, corresponding to the two transitions between canonical stance configurations. For a tetrapod gait, we would expect a three-cycle associated with the three transitions between the three tetrapod stance configurations. Correspondingly, a wave gait would produce a six-cycle corresponding to the transitions between each of the independently swinging limbs. Previous studies have noted that deviations in step timing can produce intermediate non-canonical stance configurations (*Mendes et al., 2013*). Yet, even if these intermediate configurations are present, the periodicity of the relative abundance of three-, four- and five-foot stance categories as a function of limb phase still provides information about potential underlying gait patterns.

To calculate the phase of each limb at each point in time, we decomposed the time series of the limb position in the direction parallel to the fly's body axis into amplitude and phase components using the discrete-time analytic signal method (*Figure 2—figure supplement 1*) (see Materials and methods) (*Boashash and Reilly, 1992*; *Gabor, 1946*; *Lamb and Stöckl, 2014*; *Marple, 1999*; *Vakman and Vaĭnshteĭn, 1977*). This transformation gives an estimate of the phase of each limb at each timepoint and has previously been applied to analyze cockroach limb oscillations

(*Couzin-Fuchs et al., 2015*). We then visualized the number of feet down as a function of each limb's phase to count the number of transitions per step cycle (*Figure 2F*, *Figure 2—figure supplement 2*). If canonical wave and tetrapod gaits occurred with significant frequency in our data, we would expect to see walking speeds where the number of feet down oscillates as a 6- or 3-cycle. Yet, the frequency of all limb categories peaked twice per cycle at all forward walking speeds. This two-cycle persists even when the threshold that distinguishes between swing and stance is halved, a manipulation that accentuates any effects of measurement noise. The persistent two-cycle in the data is inconsistent with the existence of tetrapod or wave gaits in a large fraction of measured trajectories. However, if tetrapod or wave gaits accounted for only a small fraction of the walking data, their characteristic periodicity might not be visible in this analysis.

## Contralateral limbs are antiphase across all walking speeds

To investigate variability in the coordination among limbs, we examined the full distributions of pairwise relative limb phases (*Figure 3A–B*). Within the set of canonical gaits, each has a characteristic set of phase offsets among the six limbs (*Collins and Stewart, 1993*). The most probable relative phase configuration in our data was not consistent with any single canonically defined gait (*Figure 3—figure supplement 1*). Like the canonical tripod gait, all contralateral pairwise relative phases maintain a mean relative phase of a half-cycle ($\langle \Delta \phi \rangle \approx 0.5$) (*Wosnitza et al., 2013*). However, unlike a perfect tripod gait, ipsilateral mid-fore and hind-mid pairings are not on average in antiphase ($\langle \Delta \phi \rangle \approx 0.4$), while hind-fore pairings are also not in phase ($\langle \Delta \phi \rangle \approx 0.85$), consistent with previous observations (*Wosnitza et al., 2013*). The observed pairwise relative phases are also inconsistent with expected phasing for canonical tetrapod and wave gaits (*Figure 3—figure supplement 1*).

To understand how phase relationships depend on forward walking speeds, we estimated the distributions of all pairwise limb relative phases conditioned on forward walking speed (*Figure 3—figure supplement 2*). Across all forward velocities, these distributions were unimodal, allowing us to characterize them by their first and second circular moments. Strikingly, at all forward walking speeds, all three contralateral limb pairings maintain a constant mean phase offset of a half-cycle (*Figure 3C*). Mean ipsilateral limb relative phases, in contrast, are not constant across walking speeds, and approach a tripod-like coupling at the fastest forward walking speeds (*Figure 3C*).

Notably, all pairwise relative phases exhibit a monotonic decrease in angular deviation as forward walking speed increases (*Figure 3D*). One interpretation for this decrease in variance is that inter-limb coupling is weaker at low speeds than at fast speeds (*Aminzare and Holmes, 2018*; *Aminzare et al., 2018*). This possibility is consistent with the observation that low-speed walking is more sensitive to sensory information and correspondingly more variable (*Berendes et al., 2016*; *Isakov et al., 2016*). A second interpretation is that some of the variability results from the characteristics of phases for waveforms with duty cycles that deviate from one-half (*Couzin-Fuchs et al., 2015*; *Lamb and Stöckl, 2014*; *Vakman and Vaĭnshteĭn, 1977*) (*Figure 3—figure supplement 3*). When measuring the relative phases between limbs, each limb sweeps out one half-cycle while in stance and another half-cycle while in swing. When the limbs have asymmetric duty cycles, they sweep out different fractions of a cycle per unit time when in swing versus stance. Thus, slow walking, which is empirically characterized by longer stances and approximately the same swing durations as fast walking, will necessarily have greater variance in relative phasing. Both explanations likely contribute to the observation of greater variance in limb relative phases at low speed.

## The joint distribution of limb relative phases is unimodal

To determine the preferred phase offsets used by the control circuits that govern walking, we examined the joint distributions of inter-limb relative phases (*Figure 3E*). If *Drosophila* used multiple distinct gaits, we would expect to see multiple peaks in this probability distribution (*Figure 3—figure supplement 1*). For instance, in this view, tripod gaits exist in one region, tetrapod gaits exist in two, and wave gaits exist in a fourth. Yet, the observed distribution has only a single peak, suggesting that the spontaneously walking fruit fly uses a single continuum of limb relative phases (*Figure 3E*). This observation holds across all forward walking speeds; in particular, there was only a single mode even at the slowest walking speeds (*Figure 3—figure supplements 2* and *3*). The most abundant relative phasing is distinct from the peaks predicted for any canonical gait; however, it is closest to that of a canonical tripod gait. There were no peaks in the distribution at locations

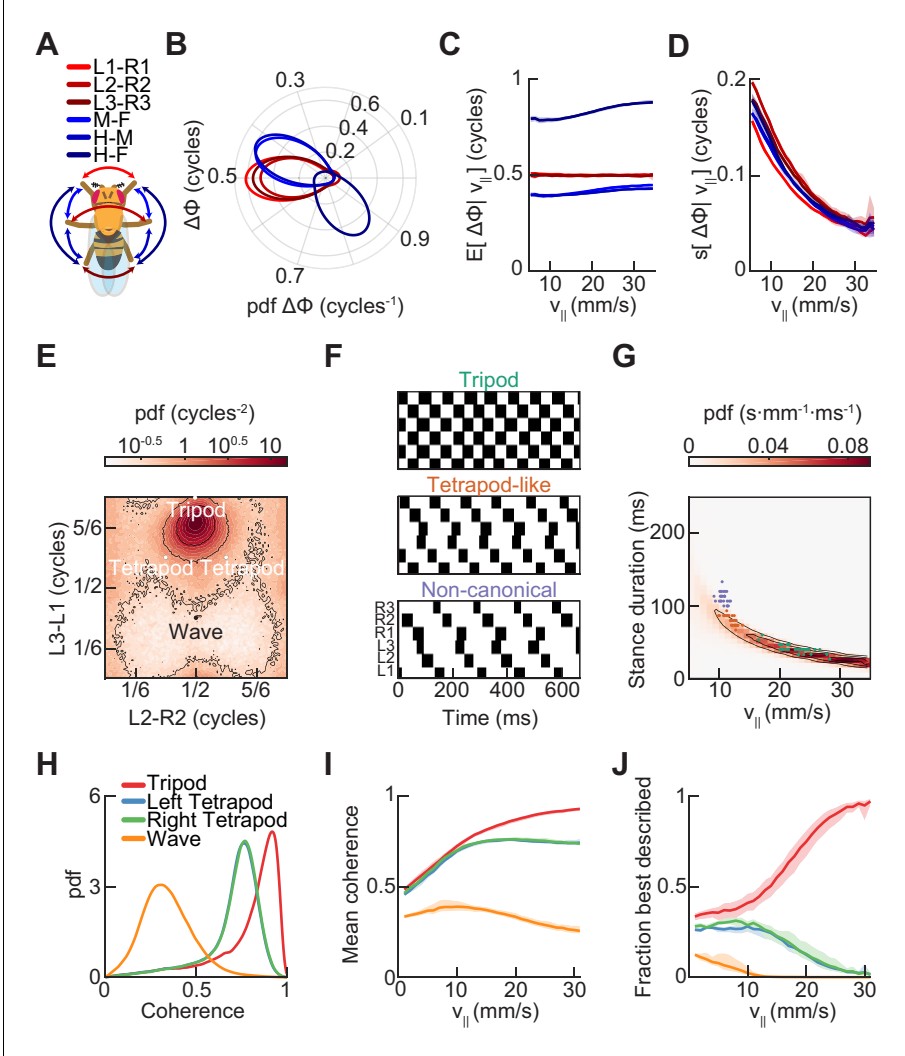

**Figure 3.** Relative phase measurements reveal a continuum of coordination patterns across all walking speeds with contralateral limbs in antiphase. (**A**) Diagram of pairwise limb relative phases, with ipsilateral pairs in *blue* and contralateral pairs in *red*. (L1-R1 = Left forelimb – Right forelimb, L2-R2 = Left midlimb – Right midlimb, L3-R3 = Left hindlimb – Right hindlimb, M-F = Midlimbs – Forelimbs, H-M = Hindlimbs – Midlimbs, H-F = Hindlimbs – Forelimbs). (**B**) Distributions of limb relative phases (Δϕ) for ipsilateral and contralateral pairs across all velocities. (**C**) Circular means of relative phases as a function of forward velocity. Contralateral relative phases are constant at 0.5 cycles across all walking speeds. Adjacent ipsilateral pairings (mid-fore and hind-mid) approach 0.5 cycles out of phase as forward velocity increases. Hind-fore ipsilateral limb pairings approach being in-phase as forward velocity increases. (**D**) Angular deviation of pairwise relative limb phases. Variance monotonically decreases in all limb pairings as forward velocity increases. (**E**) Joint probability distribution of L2-R2 and L3-L1 relative phases reveals a single peak in the distribution. (**F**) Example trajectories from freely walking fruit flies showing limb coordination patterns similar to canonical tripod (*green*) and tetrapod (*orange*), and to a non-canonical gait (*purple*) in which hind-fore limbs swing together while midlimbs swing individually. (**G**) Stance duration probability conditioned on forward velocity. All stances from example trajectories overlaid: tripod (*green*), tetrapod (*orange*), and non-canonical (*purple*). (**H**) Coherence distributions (see Materials and methods) for canonical tripod (*red*), left tetrapod (*blue*), right tetrapod (*green*), and wave gaits (*orange*). (**I**) Mean coherence at each forward walking speed for canonical tripod (*red*), left tetrapod (*blue*), right tetrapod (*green*) and wave gaits (*orange*). (**J**) Fraction of data best described by each canonical gait as a function of forward walking speed.

The online version of this article includes the following figure supplement(s) for figure 3:

**Figure supplement 1.** Synthetic canonical gaits differ in relative phasing from the coordination patterns used by free-walking *Drosophila*.

*Figure 3 continued on next page*

*Figure 3 continued*

**Figure supplement 2.** The conditional distributions of pairwise limb relative phases are unimodal at all forward walking speeds.

**Figure supplement 3.** Additional measurements of limb phases.

corresponding to canonical tetrapod or wave gait configurations. However, the tails of this distribution extend to the location of both canonical tetrapod peaks, which allowed us to identify individual trajectories with tetrapod-like characteristics.

Consistent with previous studies, we could extract individual trajectories that are similar to canonical tripod and tetrapod gaits by selecting trajectories with the expected duty cycles (*Figure 3F–G*) (*Mendes et al., 2013*; *Pereira et al., 2019*). Interestingly, the tetrapod-like examples appear to have a characteristic cross-body offset in step timing, so that the limbs that should swing together in the canonical tetrapod gait are actually slightly offset in time. This offset maintains approximate antiphase between contralateral pairs of limbs, rather than the expected phase offset of the canonical tetrapod gait (*Figure 3F*). In addition, we observed a previously unidentified non-canonical coordination pattern in which fore- and hind-limbs swing together while each mid-limb swings alone (*Figure 3F*). This pattern is distinct from all previously described canonical gaits, but might be expected when considering a single continuum of coordination patterns. In particular, this non-canonical configuration was predicted from an early continuum view of limb coordination (*Wendler, 1964*; *Wilson, 1966*).

## Template-matching suggests little evidence for preferred canonical tetrapod and wave gaits

We next sought to systematically investigate how closely our data matched the configurations of relative limb phasing specified by each of the canonical gaits. To do so, we defined a template-matching coherence metric based on a simple model for networks of coupled oscillators (see Materials and methods). The coherence falls between zero and one, with unity corresponding to an exact match to the template phase configuration. The tripod coherence distribution is sharply peaked around ~0.9, indicating a close template match, while the distributions for the other canonical gaits were broader and peaked at lower coherences (*Figure 3H*). If the fly preferentially used different gaits at different forward walking speeds, then we would expect to see a peak in the conditional coherence distribution for a given canonical gait at some characteristic forward velocity. Yet, there were not prominent peaks in our data, and all coherences yielded low average values at slow walking speeds (*Figure 3I*). At those slower speeds, where mean coherences were low, the tetrapod coherences became more likely to be the highest coherence (*Figure 3J*). This is consistent with the observation of tetrapod-like coordination patterns at low speeds (*Figure 3F–G*), and with the increased spread of the distribution of relative phases at low speeds (*Figure 3D,E*, *Figure 3—figure supplements 2* and *3*; *Wosnitza et al., 2013*). However, even when the largest coherence was not tripod, these non-tripod coherences were lower than when the tripod coherence was largest (*Figure 3—figure supplement 3*). Since the tetrapod coherence of a perfect tripod gait is $3^{-1/2} \sim 0.6$, this coherence metric may not show strong contrasts under some deformations in coordination. Therefore, under these analyses, we do not observe strong evidence for the preferred use of canonical tetrapod and wave gaits.

## A single low-dimensional manifold describes limb coordination during walking

To directly visualize the structure of the continuum of coordination patterns indicated by our phase analysis, we first applied principal component analysis to our limb kinematic data (*Figure 4—figure supplement 1*). The principal components of these data occur in degenerate, phase-shifted pairs. The projection into the first two principal components provides some information about the phasic structure of our data. However, variations in walking frequency are not well-represented by this projection. In particular, the first two principal components emphasize a single frequency, so that frequencies that are both higher and lower are mapped to the same radius in the plane. Furthermore,

the projection into the first three principal components does not provide greater insight. Therefore, principal component analysis provides some information about the structure of the data, but does not permit easy visualization of its full structure in two or three dimensions.

We therefore applied Uniform Manifold Approximation and Projection (UMAP) to generate a low-dimensional embedding of our data (*McInnes et al., 2018*). UMAP minimizes the cross-entropy between topological representations of the local distance metric on some high-dimensional data manifold and on a low-dimensional embedding manifold. Among nonlinear dimensionality reduction techniques, UMAP is ideal for this application because it attempts to preserve the local and global topology of the high-dimensional data, in contrast to algorithms like t-SNE, which is designed to emphasize local structure at the expense of global structure (*Becht et al., 2019*; *McInnes et al., 2018*).

Using UMAP, we generated low-dimensional representations of $10^5$ randomly-sampled segments of limb positional data, each with a half-window length of 100 ms (15 frames). This sampling thus had a dimensionality of 31 frames by 6 limbs by two spatial coordinates, or 372 total dimensions. Our data was represented by a bell-shaped manifold embedded in three-dimensional space (*Figure 4*). The linear dimension along the axis of the bell is parameterized by the mean stepping frequency (*Figure 4A*, *Figure 4—figure supplement 2*), while the cyclic coordinate is defined by a single global phase (*Figure 4B*, *Figure 4—figure supplement 2*). Antiphase between contralateral limbs is preserved at all phases of the global oscillator (*Figure 4B*, *Figure 4—figure supplement 3*). Embedding individual trajectories illustrates a clockwise rotation during forward walking when viewed from the mouth of the bell (*Figure 4C*). The number of feet in stance at the central timepoint of the sampled segments oscillates as a two-cycle as a function of the cyclic coordinate (*Figure 4D*). Consistent with the absence of multiple preferred configuration patterns, the density of points within this manifold along its axial dimension is unimodal (*Figure 4—figure supplement 2*). If one halves or doubles the segment length, the axial extent of the manifold is compressed or extended, intuitively corresponding to one's ability to extract frequency information from the time series, following the frequency-time uncertainty principle (*Figure 4—figure supplement 4*). Thus, this manifold learning analysis of our data reveals a structure that exactly corresponds to the intuitive representation of a system of coupled oscillators with a shared frequency that varies between segments.

To test how this UMAP visualization would act on a dataset known to contain distinct gaits, we generated a synthetic dataset in which all three canonical gaits were present, each at a continuum of stepping frequencies (see Materials and methods). To generate an embedding comparison that provided evidence for the absence of distinct gaits, one would need to have a model for the structure of gait transitions. However, there is not a clear null model for such transitions. In the absence of a null model, this analysis considers only pure canonical gaits. Even when canonical tetrapod and wave gaits each accounted for only 1% of the segments, this embedding clearly showed distinct manifolds for the canonical gaits (*Figure 4—figure supplement 5*). Therefore, although this simulation does not provide direct evidence for the absence of multiple distinct gaits, it provides additional evidence that the manifold structure of our data is consistent with a single continuum, since data containing discrete gaits could have yielded distinct manifolds for each.

## Varying one parameter in a simple model generates key attributes of observed walking patterns

Our experimental data suggest that walking speed modulation in *Drosophila* may be consistent with a simple control circuit with speed-independent coupling. We tested this possibility by constructing a minimal, rule-based generative model for phase dynamics (*Figure 5A–B*) (see Materials and methods). Conceptually, this model is related to rule-based control algorithms for walking robots, but it differs from most such models in that it generates phase dynamics rather than target limb placement positions (*Kwak and McGhee, 1989*; *Song and Waldron, 1987*; *Wettergreen and Thorpe, 1992*). In this model, we varied the stance duration while keeping the inter-limb coupling and swing duration fixed, thus generating posterior-to-anterior metachronal waves along each side of the body (*Figure 5A*). The interval between metachronal waves was governed by the stance duration, which is the single modulated parameter in the model. The two sides of the body were coupled such that they were in antiphase, independent of stance duration.

This simple construction recapitulated several key observations in our data. First, it can generate canonical tripod and wave gaits, as well as a continuum of intermediate coordination patterns

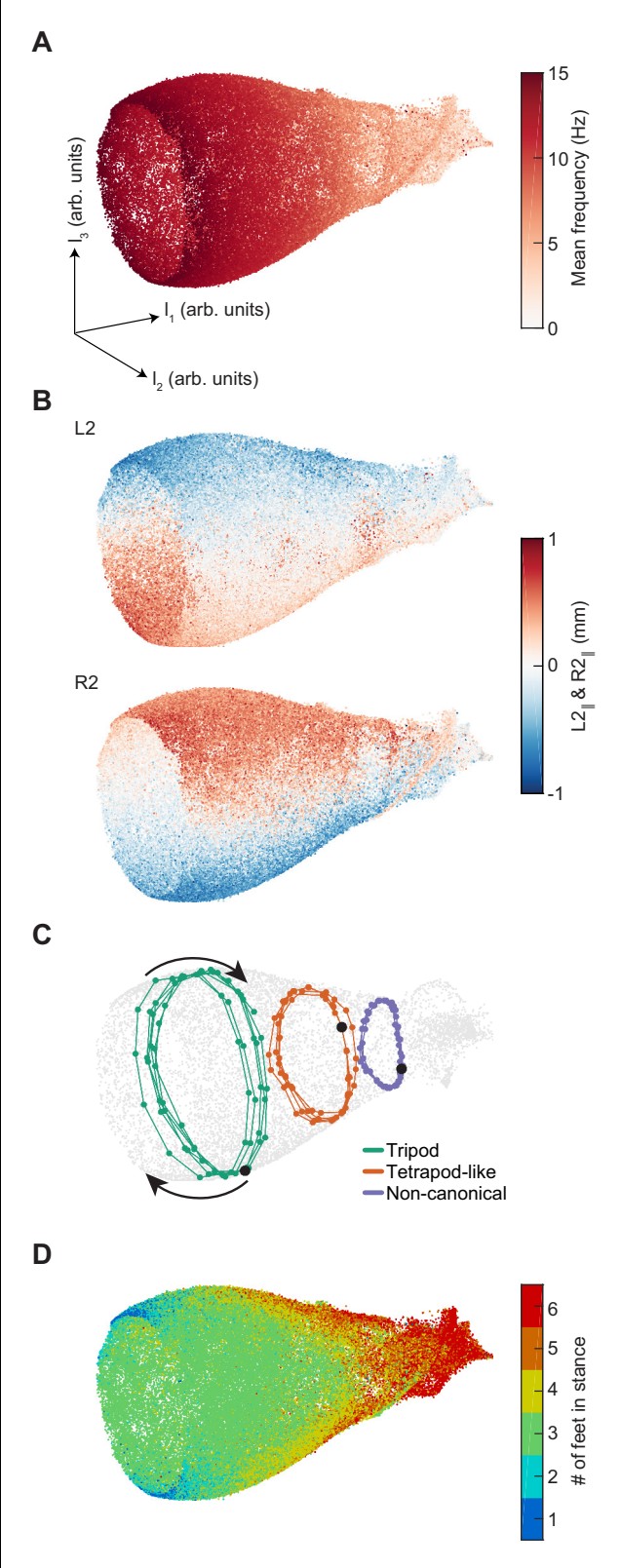

**Figure 4.** Dimensionality reduction reveals the manifold structure of limb coordination patterns. (**A**) UMAP embedding of limb coordinate time series colored by the mean frequency of forward walking. Frequency (correlated with forward walking velocity) maps to the height along the vase-shaped manifold. (**B**) Embedding colored by the mean-subtracted positions of the left and right midlimbs. The global phase of the walking behavior

*Figure 4 continued on next page*

*Figure 4 continued*

defines the location along a cross-section of the manifold. (C) Tripod (*green*), tetrapod-like (*orange*), and non-canonical (*purple*) trajectories from *Figure 3F–G* embedded in the UMAP space. Start of each trajectory is indicated by a *black* circle. Arrows indicate the trajectory direction. (D) Embedding colored by number of feet in stance. The number of feet in stance changes with a periodicity of two per cycle at all forward walking speeds. The online version of this article includes the following figure supplement(s) for figure 4:

**Figure supplement 1.** Principal component analysis of limb kinematic data.
**Figure supplement 2.** Representation of UMAP embedding in cylindrical coordinates.
**Figure supplement 3.** Contralateral antiphase is preserved at all phases of the global oscillator.
**Figure supplement 4.** Changing the segment duration dilates the axial extent of the UMAP manifold while maintaining the same structure.
**Figure supplement 5.** The manifold structure of synthetic canonical gaits differs qualitatively from that of free-walking *Drosophila*.

(*Figure 5B*). Importantly, because the phase difference across the body is always one half-cycle, this model cannot produce the canonical tetrapod gait, but it can produce the tetrapod-like pattern with a characteristic offset in swings observed in our experimental data. With an appropriate choice of stance duration, it also generated the observed non-canonical limb pattern in which fore- and hind-limbs swing simultaneously while mid-limbs swing individually. With this model, the number of feet in stance phase varied as a function of the forward walking speed, as observed in experimental data (*Figure 5C*). The model also reproduced the measured two-cycle in the number of feet down as a function of limb phase (*Figure 5D*, *Figure 5—figure supplement 1*). This two-cycle occurs in the model because each ipsilateral posterior-to-anterior wave is effectively a single event, and they are always exactly out of phase. The mean contralateral relative phases are constant across forward walking speeds, while the mean ipsilateral relative phases vary in a speed-dependent manner, consistent with our experiments (*Figure 5E*). Finally, this model recapitulates the decrease in variance in the relative phases with increasing forward speed observed in our data (*Figure 5F*). Consistent with experimental observations, embedding the model-generated data using UMAP produces a single manifold (*Figure 5G*). Overall, this simple model with speed-independent limb coupling and a single control parameter qualitatively captures major characteristics of the symmetric variability in *Drosophila* spontaneous walking coordination.

## Turns are achieved through asymmetric, segment-specific modulations of limb movements

Our simple model shows that symmetric variability in limb movements can be described using a single parameter to control speed. Turning is inherently a form of asymmetric variation, as it requires that the outside legs of the fly traverse a greater distance than the inside legs. This path length differential could be achieved by modulating any combination of three parameters of limb kinematics: the frequency of stepping, the length of steps, and the direction of steps. We therefore sought to determine which of these parameters are modulated during turning. To exclude the effect of forward speed modulation on these parameters, we restricted our analysis to forward speeds between 15 and 20 mm/s, the range with the greatest quantity of turning data.

On average, the swing durations of the inside mid- and hind-limbs decrease as a function of turning rate, while their stance durations increase (*Figure 6A–B*). The swing and stance durations of the remaining limbs are oppositely modulated (*Figure 6A–B*). At the highest yaw rates, the net modulation of stepping frequency is about 25%. The step length of the inside forelimb is not significantly modulated during turning (*Figure 6C*, *Figure 6—figure supplement 1*). The step lengths of the outside limbs increase with yaw rate while step lengths of the inside midlimb and hindlimb decrease with increasing turning rate. Overall, the observed limb-specific modulations of step frequency and step length are consistent with reports in other hexapods (*Franklin et al., 1981*; *Frantsevich and Mokrushov, 1980*; *Graham, 1972*; *Gruhn et al., 2016*; *Jindrich and Full, 1999*; *Strauß and Heisenberg, 1990*; *Zollikofer, 1994*; *Zolotov et al., 1975*).

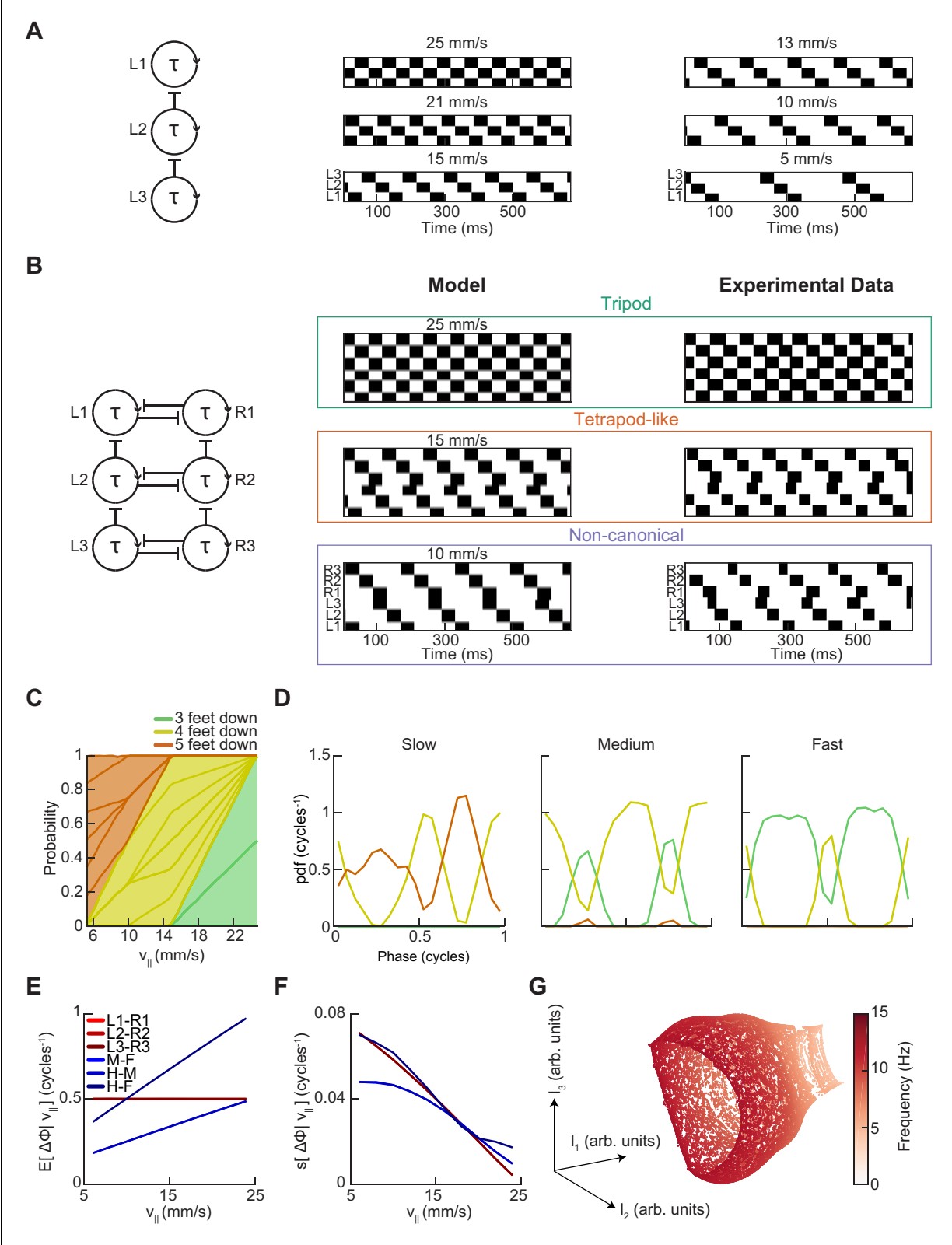

**Figure 5.** A single-parameter model with speed-independent coupling predicts a continuum of inter-limb coordination patterns. (A) A single-parameter phase oscillator model to generate metachronal waves (see Materials and methods). The frequency of metachronal waves is determined by a single parameter, $\tau_{stance}$. Limb oscillators are coupled by constant unidirectional inhibition from posterior limbs to anterior limbs. (B) A minimal model of spontaneous walking in the fruit fly may be obtained by linking two of the units diagrammed in (A) with bidirectional contralateral coupling. Forward

*Figure 5 continued on next page*

*Figure 5 continued*

walking speed is determined solely by stance duration. Example swing-stance plots illustrate the ability to generate walking patterns similar to canonical tripod ($v_{||}$=25 mm/s) and tetrapod-like gaits ($v_{||}$=15 mm/s). Additionally, this model generates the non-canonical limb coordination pattern identified in our data ($v_{||}$=10 mm/s), in which midlimbs swing independently while contralateral fore and hind limbs swing together. As observed in our experimental data, the tetrapod-like pattern generated by this model deviates from canonical tetrapod as relative phases between contralateral limbs are antiphase. Example trajectories from *Figure 3F* are presented for comparison to model trajectories. This model was used to generate the data analyzed in panels (C–G). (C) Relative probability of the number of feet in stance as a function of forward walking speed. As walking speed increases, the number of feet in stance decreases. (D) Relative probability of number of feet in stance as a function of midlimb phase for bottom, middle, and top thirds of forward velocity distribution. The number of feet down oscillates with a frequency of two per single-limb-stride. (E) Circular means of limb relative phases for each pairwise relationship as a function of forward velocity. Consistent with experimental data, contralateral relative phases are constant at ½ cycles across all walking speeds. Adjacent ipsilateral pairings (mid-fore and hind-mid) approach antiphase as walking frequency (forward velocity) increases. Hind-fore ipsilateral limb pairings approach being in-phase as forward velocity increases. (F) Angular deviation of pairwise relative limb phases. Variance of relative phases decreases in all limb pairings as forward velocity increases. (G) UMAP embedding of model-generated trajectories. All trajectories lie on a single manifold.

The online version of this article includes the following figure supplement(s) for figure 5:

**Figure supplement 1.** A single-parameter model generates a two-cycle coordination pattern across all walking speeds.

The remaining kinematic parameter is the direction of limb movement during stance, which can also change the total path traversed by each limb (*Figure 6D*, *Figure 6—figure supplement 1*).

Our measurements of the two forelimbs show nearly identical modulations of stepping frequency and modest differences in step length modulations. The remaining path length differential between forelimbs is achieved through modulation of the stance direction of the inside forelimb, which is modulated more dramatically than that of other limbs, by up to about 45°. This suggests that the fly achieves the required difference in forelimb path length by directing forelimb motions away from, rather than along, its path. This is consistent with measurements in cockroaches (*Jindrich and Full, 1999*; *Mu and Ritzmann, 2005*) and in tethered stick insects (*Gruhn et al., 2009*). Thus, unlike forward walking speed changes, which are dominated by global modulations of a single parameter, turns are achieved through asymmetric and limb-specific modulations of multiple parameters of limb movement.

## Turns are aligned to preferred phases of the limb oscillator

Next, we sought to understand whether symmetric and asymmetric patterns of limb movement are coupled. In particular, we examined whether flies execute turns at all phases of their walking pattern or if turns occur at particular limb configurations. To visualize the gross placement of limbs during turning, we estimated the spatial distributions of limb positions over all walking behaviors (*Figure 7A*) and at yaw rate extrema (*Figure 7B*). These distributions demonstrate that turns occur at a preferred spatial configuration of the walking oscillator. To understand how turns are timed relative to the underlying straight-walking coordination pattern, we examined the distributions of the instantaneous phases of each of the limbs at yaw rate extrema (see Materials and methods). Turns were preferentially aligned to a particular phase for each leg, which corresponds to a preferred phase of the walking oscillator (*Figure 7C–D*). In particular, there exists a statistically significant difference between the distributions of phases for all timepoints and for yaw extrema ($p < 10^{-5}$ for all limbs, see Materials and methods, *Table 4*). Thus, turns consist of precisely-timed modulations of the underlying walking coordination pattern, which couple the asymmetric variability to the symmetric coordination patterns.

## Perturbations of forward walking speed modulate stance duration

In the analyses presented so far, we have characterized the structure of variability in spontaneous walking behavior. We next sought to test whether the structure of evoked behavior was similar to spontaneous behavior. Our experiments and model both suggest that stance duration controls forward walking speed during spontaneous behavior in *Drosophila*. Since flies also transiently modulate their walking speed in response to stimuli, we wondered whether this transient modulation was along the same dimension as the spontaneous behavior. For instance, while spontaneous walking speed is regulated primarily by stance duration, transient modulations of walking speed could

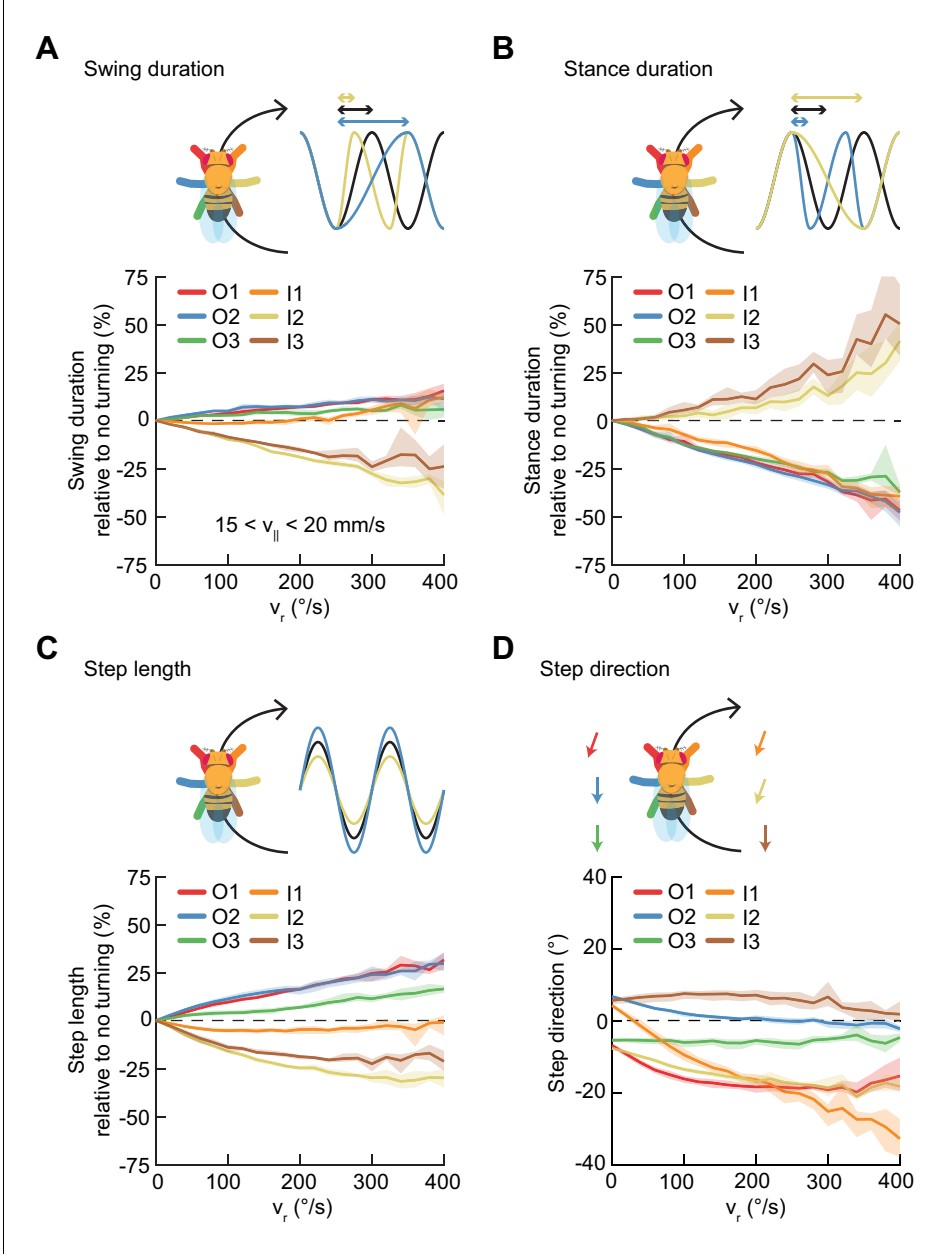

**Figure 6.** Asymmetric, segment-specific modulations of limb movement underlie turning. Error patches show 95% confidence intervals of the mean obtained from bootstrap distributions over experiments (N = 8 videos; see Materials and methods). (A) Modulation of swing duration in individual limbs relative to straight walking as a function of symmetrized yaw velocity. *Red*, *blue* and *green* indicate the outside fore-, mid-, and hindlimbs respectively. *Orange*, *yellow* and *brown* indicate the inside fore-, mid-, and hindlimbs respectively. To exclude the effect of forward speed modulation on limb movement parameters, the analysis is restricted to forward velocities between 15 and 20 mm/s. The swing durations of the inside mid- and hindlimbs decrease with increasing yaw rate while those of the remaining limbs increase slightly with yaw rate. (B) As in (A), but for stance duration. The stance durations of the inside mid- and hindlimbs increase with increasing yaw rate while those of the remaining limbs decrease with yaw rate. (C) As in (A), but for step length. The step lengths of the outside limbs increase with yaw rate, while those of the inside mid- and hindlimbs decrease with yaw rate and that of the inside forelimb is barely modulated. (D) As in (A), but for step direction. The direction of steps in the inside and outside forelimbs is shifted such that their movements are directed outwards relative to the curvature of its path.

The online version of this article includes the following figure supplement(s) for figure 6:

**Figure supplement 1.** Average modulations of limb movement parameters.

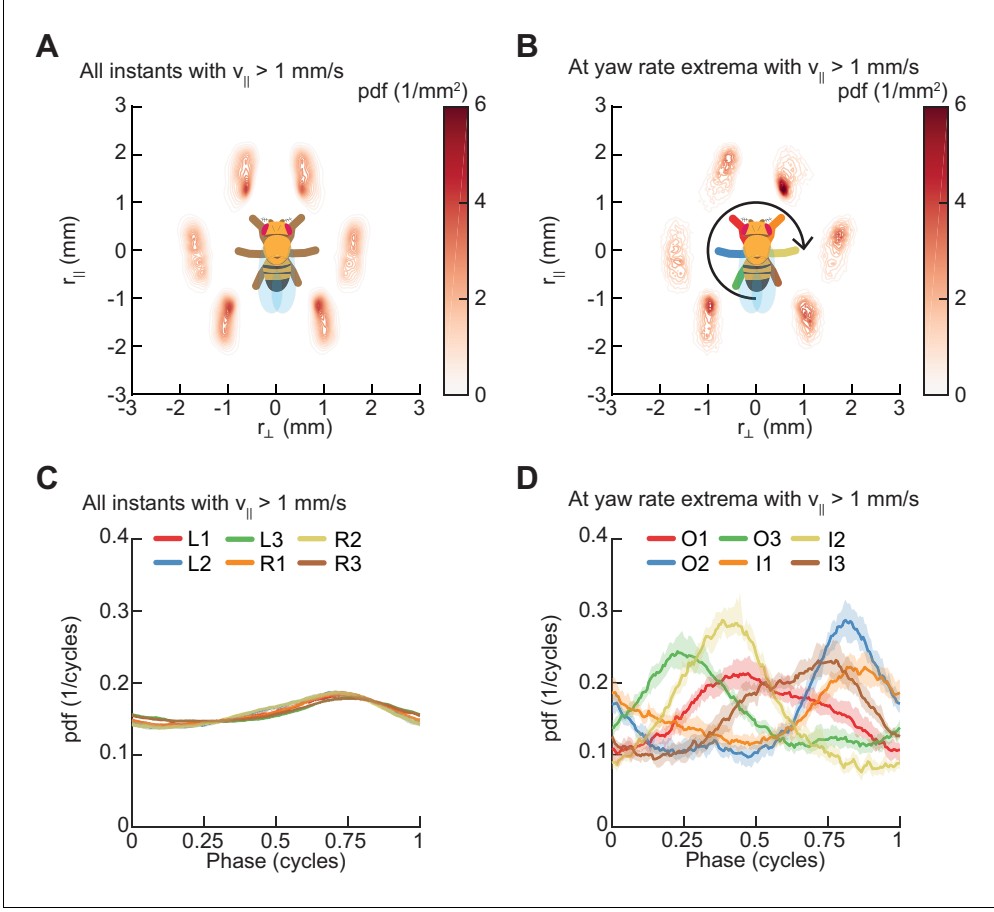

**Figure 7.** Spontaneous turns are aligned to preferred phases of the limb oscillator. Error patches show 95% confidence intervals obtained from bootstrap distributions over experiments (N = 8 videos; see Materials and methods). (**A**) Probability density function of limb positions in the egocentric frame of the fly over all walking behaviors. (**B**) As in (**A**), but at yaw rate extrema. The tripod containing the inside forelimb is in nearly its most posterior position, while the tripod containing the outside forelimb is in nearly its most anterior position. (**C**) Circular probability density functions of the instantaneous phases of each limb over all walking behaviors. Phases between ½ and 1, corresponding to stance phase, are more probable than phases between 0 and ½, corresponding to swing phase, due to the fact that the duration of stance phase is on average greater than that during swing phase. (**D**) As in (**C**), but at yaw rate extrema. There exists a preferred phase of each limb at which yaw rate extrema occur, which differs significantly from the corresponding time-invariant distribution (see Materials and methods).

instead be modulated by changing the step length, swing duration, or both. To investigate these possibilities, we generated transient slowing in walking flies both by directly activating a set of command neurons and by presenting visual stimuli that induced slowing.

Previous work identified a set of command neurons in a pathway named moonwalker, which regulates the speed and direction of walking (*Bidaye et al., 2014*). When these neurons are tonically activated, the fly walks backwards. We expressed Chrimson in these neurons and then stimulated freely-walking flies with 8 ms pulses of red light (see Materials and methods, *Table 1*) (*Klapoetke et al., 2014*). Short-timescale activation of moonwalker neurons caused a transient reduction in forward walking speed that lasted ~100 ms (*Figure 8A*). This slowing is consistent with the suppression of forward walking elicited by activating a subset of the moonwalker neurons (*Bidaye et al., 2014*). This reduction in forward walking speed was accompanied by a transient increase in stance duration, the control parameter for speed regulation in our coordination model (*Figure 8B*). In individual trials, all limbs increased their stance duration over hundreds of milliseconds in response to the brief neural activation (*Figure 8C*). When visualized using UMAP, moonwalker activation corresponded to a

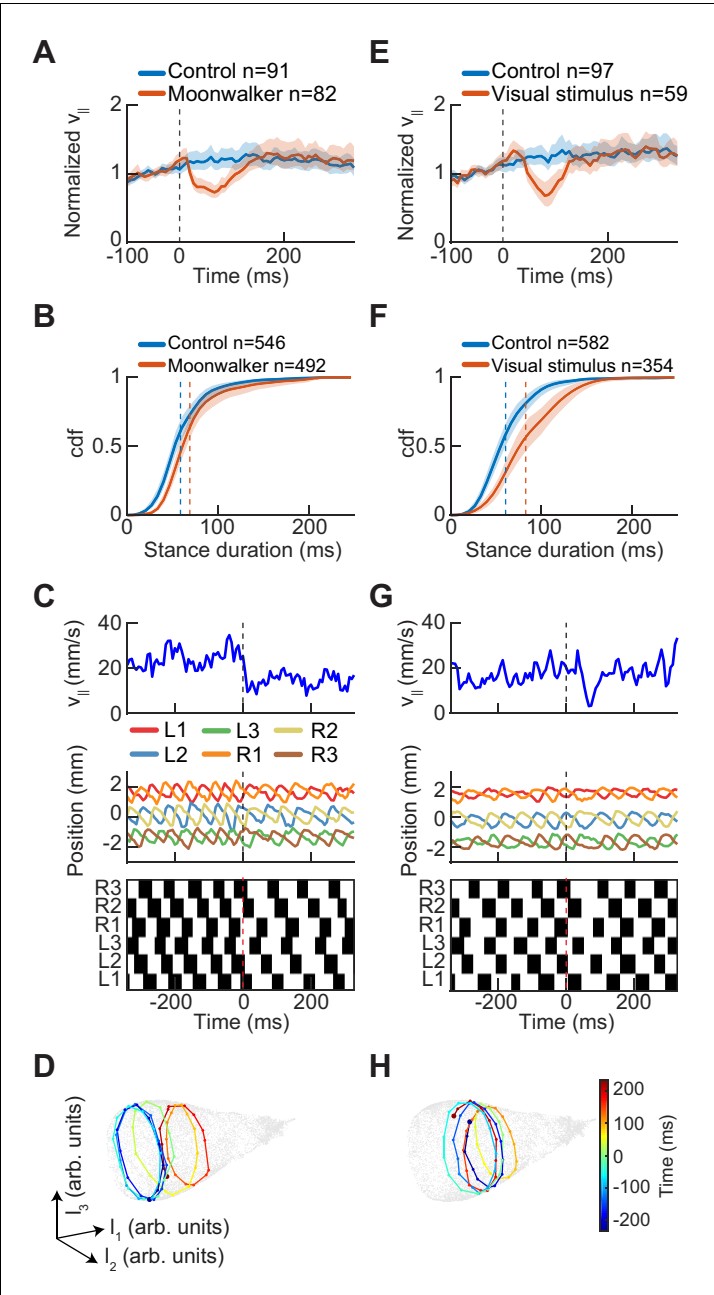

**Figure 8.** Experimental perturbations of walking speed modulate stance duration. (**A**) *Drosophila* normalized forward speed over time in response to 8 ms optogenetic activation of moonwalker neurons (*red*, n = 82) versus random trigger control (*blue*, n = 91). (**B**) Cumulative distribution function of first stance duration post perturbation for all limbs for moonwalker manipulation (*red*, n = 492) versus random trigger control (*blue*, n = 546). Includes stance durations from all six limbs. Distributions are significantly different as measured by a two-sample Kolmogorov-Smirnov test (N = 91, 82; p<0.05; D = 0.1965). (**C**) Single trial of walking over time in response to moonwalker activation. Forward walking speed (*blue*) over time. Limb positions in the direction parallel to the fly's body axis over time. Step plot for the fly over time with swing (*black*) and stance (*white*) in each of the limbs. Post manipulation, stance duration increases in all limbs. (**D**) The trajectory of a single trial in the UMAP embedding space shows that moonwalker activation induces shifts along the length of the manifold. The time relative to activation onset at the center of each embedded trajectory is indicated by color, with *dark blue* corresponding to times before onset ($-200 \leq t \leq 0$ ms), *green* corresponding to the onset of activation ($t = 0$ ms), and *dark red* corresponding to times after activation ($0 \leq t \leq 200$ ms). (**E**) *Drosophila* normalized forward speed over time in response to 50 ms translation of random dot stimuli moving at 720, 960, or 1440 °/s (*red*, n = 59) versus random

*Figure 8 continued on next page*

*Figure 8 continued*

trigger control (*blue*, n = 97). (**F**) Cumulative distribution function of first stance duration post perturbation for all limbs for visual stimulus manipulation (*red*, n = 354) versus random trigger control (*blue*, n = 582). Includes stance durations from all six limbs. Distributions are significantly different as measured by a two-sample Kolmogorov-Smirnov test (N = 97, 59; p<0.05; D = 0.2834). (**G**) As in (**C**), but for visual stimulation. (**H**) As in (**D**), but for visual stimulation.

deflection along the stepping frequency dimension of the manifold, consistent with a modulation of stance duration (*Figure 8D*). Thus, symmetrically activating a small subset of command neurons shifts the fly's limb coordination along the same manifold as spontaneous symmetric variation.

Last, we sought to test whether more naturalistic slowing commands also modulate stance duration. Previous studies in tethered and freely-walking flies have shown that a variety of visual stimuli evoke slowing (*Creamer et al., 2018*; *Katsov and Clandinin, 2008*; *Silies et al., 2013*). We presented rigidly translating random dot patterns above freely-walking flies. In response to this visual stimulus, flies slowed transiently, consistent with previous studies (*Figure 8E*). As with the moon-walker manipulation, visual stimulation produced a transient increase in stance durations (*Figure 8F–G*). The visual stimulus-based perturbations also induced shifts along the frequency dimension of the UMAP manifold, corresponding to the change in stance duration (*Figure 8H*). Therefore, slowing evoked by visual stimuli generated transient changes along the same manifold as the spontaneous symmetric modulations that control walking speed.

## Discussion

In this study, we comprehensively characterized the variability in *Drosophila* limb coordination during spontaneous walking behaviors. To fully sample the space of limb coordination patterns, we developed an automated method for video annotation (*Figure 1*). We investigated the distribution of limb coordination patterns across forward walking speeds in freely-moving flies (*Figures 2* and *3*). Our experimental data lie on a single manifold, consistent with a continuum of coordination patterns (*Figure 4*). A simple model with fixed coupling could reproduce the observed symmetric patterns of walking at a wide range of speeds by varying only stance duration (*Figure 5*). In contrast, asymmetric variability associated with turning is characterized by small, precisely-timed, and limb-specific modulations of this underlying straight-walking coordination pattern (*Figures 6* and *7*). Evoked changes in walking speed generated changes in limb coordination similar to the structure of spontaneous modulations in walking speed (*Figure 8*).

### A single continuum of leg coordination patterns suggests a simple neural control circuit

A variety of computational models have been developed to describe patterns of hexapod locomotion (*Aminzare and Holmes, 2018*; *Aminzare et al., 2018*; *Collins and Stewart, 1993*; *Collins and Stewart, 1994*; *Cruse, 1979*; *Cruse, 1980*; *Cruse, 1990*; *Cruse et al., 1995*; *Delcomyn, 1999*; *Graham, 1977*; *Ijspeert, 2008*). A central component of these models is the nature of inter-limb coupling, which in turn determines the predicted limb coordination patterns. Our simple model suggests that one might expect to find mutual inhibitory coupling between contralateral neuropil of the ventral nerve cord (VNC) and posterior to anterior inhibitory coupling between ipsilateral neuropil. Importantly, neither contralateral nor ipsilateral couplings need vary with walking speed. If a single continuum can describe fly walking, then models of the circuit controlling walking may be made substantially simpler, since they need not account for multiple distinct coordination patterns. Instead, a simpler control circuit need only be able to vary stance duration while maintaining ipsilateral and contralateral coupling. This suggests the intriguing possibility that walking *Drosophila* may use a qualitatively different locomotor control circuit than previously studied animals in which distinct gaits have been observed (*Alexander, 1989*; *Alexander and Jayes, 1980*; *Alexander and Jayes, 1983*; *Brett and Sutherland, 1965*; *Rayner et al., 1986*; *Srinivasan and Ruina, 2006*).

With a simple, one-parameter model, we recapitulated several major elements of the symmetric variability in spontaneous walking (*Figure 5*). However, there were aspects of our experimental data that were not captured by our simple model. Though the ipsilateral relative phases exhibited qualitatively similar speed-dependent shifts, the magnitude of these shifts is larger in our model than in our data, with greater deviations at slow forward walking speeds. There are several potential explanations for this larger range. First, our experimental data included turning, which our model does not consider. Second, the relative timing of ipsilateral swings was less variable in our model than in our experimental data. Third, each swing event in the model was of constant duration while our experimentally-measured swings were more variable in duration overall and shorter on average at slow forward walking speeds. Taken together, these differences may account for the deviations in ipsilateral phasing. These differences do not diminish the utility of this simple model, since its purpose is to illustrate the sufficiency of a single set of couplings to describe the symmetric variability in spontaneous limb coordination.

## Optimality and expected gait transitions

Prior studies have suggested that the inter-limb coordination patterns used by insects reflect an optimization against physical constraints (*Nishii, 2000*; *Szczecinski et al., 2018*). Theoretical work has suggested that continua of coordination patterns, like those identified here in *Drosophila*, minimize the energetic cost of transport when ground reaction forces are balanced across limbs (*Nishii, 2000*). In contrast, distinct gaits are optimal when non-uniform ground reaction forces are necessary to balance the body (*Nishii, 2000*). The magnitude and robustness of mechanical stability during walking may also influence the presence or absence of distinct gaits (*Szczecinski et al., 2018*). In addition to these mechanical considerations, we suggest that the simple control of the observed continuum of coordination patterns might make it preferable in animals with small circuits for controlling limbs.

## Turning limb kinematics represent small modifications of straight walking

The asymmetric variability in limb kinematics associated with turning represents a minor modification of the coordination patterns used during forward walking, with ~25% modulations of limb movement parameters at the highest yaw rates. Detailed neuromechanical and phase-reduced models in cockroaches (*Kukillaya and Holmes, 2009*; *Kukillaya et al., 2009*; *Proctor and Holmes, 2018*; *Proctor et al., 2010*) and simplified bipedal models (*Proctor and Holmes, 2008*) have shown that ~20% modulations in forelimb placement parameters can generate 90° degree turns within three strides. These theoretical findings are consistent with experimental results in cockroaches, which show that limb kinematics and dynamics during turning represent a minor modification of the symmetric coordination pattern (*Jindrich and Full, 1999*). The small magnitude of these modulations suggests a conceptual parallel between turning during walking and during flight. In flying *Drosophila*, detailed kinematic and dynamical measurements and modeling have shown that yaw is regulated through small modulations of wing rotation angle while gross wingbeat movements remain mostly unaffected (*Dickinson and Muijres, 2016*; *Fry et al., 2003*; *Lindsay et al., 2017*; *Muijres et al., 2015*).

## Distinct control mechanisms imply distinct circuit bases for yaw and speed regulation

Mechanically, turning could be achieved through asymmetric modulations of limb movement that do not affect the coordination between ipsilateral limbs (*Proctor and Holmes, 2008*). In this case, the command signals regulating turning could be unilateral but not segment-specific. However, swing duration, stance duration, step length, and step direction are all modulated on an individual-limb basis during turning. This suggests that the motor output of individual limbs in freely-walking flies is regulated during turning at the hemi-segmental level by control signals, a conclusion which is consistent with electrophysiological measurements and pharmacological manipulations in dissected stick insects (*Gruhn et al., 2016*). In contrast to the limb-specific modulation of many kinematic degrees of freedom during turning, forward speed changes are associated primarily with modulations in stance duration that are comparable across all limbs (*Figure 6*). The fact that limb movements are

modulated along different dimensions for yaw and speed control suggests differentiation between the circuit bases of symmetric and asymmetric variability beyond the simple requirement of asymmetric limb movement in turning. However, as turns are precisely timed relative to the walking coordination pattern (*Figure 7*), the control of these symmetric and asymmetric modulations must be neurally or mechanically coupled.

### Differentiated roles for the posterior and anterior neuropils of the VNC

Our detailed characterization of walking suggests that, surprisingly, the observed complexity in symmetric interlimb coordination can be reproduced with fixed, unidirectional posterior-to-anterior coupling between limbs. This model implies that the most posterior of the limb neuropils could have a differentiated role in walking speed regulation, as swing events originate in these posterior segments. Anatomical evidence supports this differentiated role. In particular, the moonwalker ascending neurons receive dendritic input almost exclusively from the metathoracic neuromere, with axons projecting to more anterior neuropils in the thorax and to the subesophageal ganglion (*Bidaye et al., 2014*). Their activation causes the fly to slow its walking (*Bidaye et al., 2014*), and they appear most active when all six limbs are in stance phase (*Chen et al., 2018*). In contrast, our kinematic measurements suggest that the forelimbs, and thus the anterior neuropil, play a distinct role in spontaneous turning, a result consistent with dynamical measurements in cockroaches (*Jindrich and Full, 1999*). Symmetry arguments suggest that walking speed and turning must be regulated through different coordinated patterns of neural activity. The difference in the roles of anterior and posterior neuropils in the two locomotor behaviors supports a hypothesis that these behaviors employ distinct circuitry.

### Separation of action selection and motor implementation in insects

Despite the phasic relationships between limbs during locomotion and grooming, a variety of studies have demonstrated that tonic activation of descending neurons (DNs) suffices to generate complex, structured motor patterns in the fruit fly (*Bidaye et al., 2014*; *Cande et al., 2018*; *Seeds et al., 2014*). Interestingly, previous electrophysiological recordings in orthopteran insects showed that neurons projecting from the insect brain to the thoracic ganglia are tonically active in all regions except for those emanating from the subesophageal ganglion (*Heinrich, 2002*). This organization suggests that the control of motor behaviors in insects physically segregates action selection and the low-level implementation of motor behaviors into the brain and VNC respectively. The manifold structure of limb coordination is compatible with tonic descending inputs that modulate limb movements, while local circuits in the VNC coordinate limbs.

These experiments have shown that variability in walking coordination patterns exists on a continuous manifold. In *Drosophila*, explorations of VNC circuits are increasingly made possible by automated behavioral tracking (*Branson et al., 2009*; *Mathis et al., 2018*; *Pereira et al., 2019*; *Williamson et al., 2018*), by preparations to visualize neural activity in the VNC (*Chen et al., 2018*; *Tuthill and Wilson, 2016*), and by genetic tools to target neurons in the VNC (*Cande et al., 2018*; *Mamiya et al., 2018*; *Namiki et al., 2018*) and to manipulate neural activity (*Luo et al., 2018*). The manifold structure of limb coordination in *Drosophila* provides a framework for dissecting the circuits that regulate walking.

## Materials and methods

### Fly strains and husbandry

Female *Drosophila melanogaster* used in experiments were grown at 25℃ on a 12 hr/12 hr light/dark cycle. Wild-type flies were staged on $CO_2$ 12–24 hr after eclosion, and run after 24 hr of staging. Flies used in optogenetic experiments were grown at 25℃ and staged 12–24 hr after eclosion on $CO_2$. When staged, flies used in optogenetic experiments were transferred to food supplemented with all-trans-retinal (ATR) following previous protocols (*de Vries and Clandinin, 2013*). Flies remained on ATR-supplemented food for 4 days prior to behavioral experiments. In all cases, flies were grown at near 50% relative humidity, and experiments were performed at 50% relative humidity.

## Experimental setup

Behavioral experiments were performed in an illuminated 5 cm diameter circular planar arena consisting of two plates of glass separated by 2.5 mm. The top glass plate was coated with Rain-X (wax) to prevent flies from walking on this surface during experiments. Above the arena, we mounted a diffusing screen and a 530 nm green Luxeon SP-01-G4 LED (Quadica Developments Inc, Lethbridge, AB, Canada) that provided background illumination for the arena (~1 µW/mm$^2$). Flies were recorded from below at 150 fps using a Point Grey Flea3 FL3-U3-13Y3M-C camera (FLIR Systems, Wilsonville, OR). The camera was positioned such that its field of view (2.2 × 2.75 cm) was approximately centered within the fly arena. All experiments were performed at 34°C to increase the frequency of fly walking (*Soto-Padilla et al., 2018*).

During walking experiments, groups of 12–15 female flies were loaded into the arena. We allowed the flies to acclimate to the arena for 20 min prior to image acquisition. We then recorded the spontaneous activity of flies in the arena for 1.1 hr. Data from eight separate recordings, with a total of 114 flies, were merged to generate the aggregate wild type, free-walking dataset analyzed in this manuscript.

## Feature extraction

To track freely-walking flies in an open arena, we decomposed the tracking process into distinct steps of centroid and footfall tracking. Our body tracking method relies on the roughly stereotyped size of fruit flies. To track the body, we binarized camera frames to separate out the darker fly pixels from the lighter background pixels. In this binarized camera frame, we identified body ellipsoids restricting our analysis to regions with roughly the correct area and eccentricity (*Bradski, 2000*). After aligning these ellipsoids, we then extracted stereotyped images of individual flies oriented such that the fly was aligned across all frames. We smoothed estimates of velocities by zero-phase filtering with a Gaussian kernel with a standard deviation of 20 ms (three frames), which eliminated this oscillation (*Figure 1—figure supplement 2*) (*Katsov and Clandinin, 2008*; *Katsov et al., 2017*). Due to the height of the arena and the temporal resolution of our camera, we were, in most circumstances, able to distinguish individual flies, even when they were in close proximity. To end trajectories when fly bodies touched and became difficult to distinguish with our algorithm, we included a threshold on the maximum ellipsoid size.

Footfall tracking was performed using linear regression. To train our footfall prediction model, we manually labeled 5,000 randomly sampled oriented fly frames output by our body tracking algorithm. Exploiting the bilateral symmetry of the fly, we augmented our training data by appending the mirror symmetric images and corresponding manual labels. We used principal component analysis to reduce the dimensionality of the image space from the full number of pixels (19,881) to 1000 (*Turk and Pentland, 1991*). This reduction in the number of model parameters was important because we wished to train our model with a limited set of hand-annotated frames. We then trained 12 separate linear regression models to independently predict each footfall location variable (*Pedregosa et al., 2011*). To ensure that correlations between the positions of limbs did not influence the predictions made for an individual limb, we masked input variables to exclude portions of the image outside of the target limb's range of motion prior to training (*Figure 1—figure supplement 1*). We fit our model with 5000 frames of annotated data, and used 10-fold cross-validation to test for overfitting. This showed that the mean Euclidean distance between predicted and hand annotated limb positions was 0.15 mm (3.5 pixels), or about 10% of stride length (*Figure 1—figure supplement 1*). This error distance is comparable to that reported for distal limb positions by published deep neural network methods for pose estimation (*Mathis et al., 2018*; *Pereira et al., 2019*).

## Fit of functional relationship between forward velocity and stance duration (*Figure 1*)

To compare our measurements of the relationship between forward velocity and stance duration with previous studies (*Mendes et al., 2013*; *Szczecinski et al., 2018*; *Wosnitza et al., 2013*), we fit a power law of the form:

$$\tau_{\text{stance}} = a \left( \frac{v_{\parallel}}{v_0} \right)^b \tag{1}$$

**Table 2.** Phase templates for canonical gait coherence analysis (***Collins and Stewart, 1993***).
Limb ordering is (L1, L2, L3, R1, R2, R3).

| Canonical gait | Phase template (cycles) |
| --- | --- |
| Tripod | [0, 1/2, 0, 1/2, 0, 1/2] |
| Left tetrapod | [1/3, 2/3, 0, 0, 1/3, 2/3] |
| Right tetrapod | [2/3, 0, 1/3, 0, 1/3, 2/3] |
| Wave | [1/6, 1/3, 1/2, 2/3, 5/6, 0] |

where $v_0 = 1$ mm/s is a fixed scaling factor used to nondimensionalize the velocity (hence $a$ has units of milliseconds). We fit this power law using the nonlinear least-squares solver built into MATLAB in the fit function, giving $a = 932.8$ ms and $b = -1.025$ with a coefficient of determination of $R^2 = 0.59$.

## Swing and stance determination (*Figures 1–4*, *6* and *8*)

Movement in the camera frame was used to determine swing and stance in each of the six limbs. Frames were categorized as stance if the smoothed instantaneous velocity of the limb was less than 20 mm/s (3.1 pixels per frame), approximately matching our error in positional measurement. Frames above this threshold were categorized as swing. Smoothing of limb velocities was performed using a five-frame moving average filter. Both thresholds were chosen manually in order to give a reasonable representation in the presence of limb position noise.

## Estimation of limb oscillator phase (*Figures 2*, *3*, *5* and *7*)

To estimate the phases of the limbs, we used the discrete-time analytic signal method. Briefly, this method uses the Hilbert transform to construct the harmonic conjugate of a real-valued signal, and then estimates the phase as the argument of the resulting complex-analytic function. In continuous time, the Hilbert transform of a real signal $s(t)$ is related to the Fourier transform as

$$H\{s\}(t) = \mathfrak{F}^{-1}\{-i \operatorname{sgn}(\xi) \, \mathfrak{F}\{s\}(\xi)\}(t) \tag{2}$$

which allows it to be defined in discrete-time by replacing the continuous-time Fourier transform by its discrete-time analog. Using the Hilbert transform, one may define the analytic signal as:

$$s_a(t) = s(t) + i \, H\{s\}(t) \tag{3}$$

From the analytic signal, the instantaneous phase is defined as

$$\phi(t) \equiv \arg\{s_a(t)\} \tag{4}$$

where the argument function is defined such that the instantaneous phase is a continuous function of time (***Vakman and Vaĭnshteĭn, 1977***; ***Boashash and Reilly, 1992***; ***Gabor, 1946***). We computed the analytic representations of the positions of each limb in the direction parallel to the fly's body axis using the *hilbert* function in Matlab (Mathworks, Natick, MA, USA). This function implements the fast-Fourier-transform-based algorithm introduced in ***Marple (1999)***. The zero-point of the resulting phase estimate corresponds to the maximally posterior point of the limb cycle. The discrete-time analytic signal method was previously used to estimate the phases of cockroach limb oscillations (***Couzin-Fuchs et al., 2015***), where it was shown that this method produced comparable results to the phase estimation algorithm introduced in ***Revzen and Guckenheimer (2008)***. We smoothed the resulting phase estimates using a third-order Savitzky-Golay smoothing filter with a window length of 15 frames, and estimated instantaneous frequencies using a corresponding differentiating filter (***Savitzky and Golay, 1964***). To calculate the relative phase differences between limbs, instantaneous phase measurements were subtracted from each other at each frame and expressed in fractions of a limb cycle.

## Coherence analyses (*Figure 3*)

To measure how well our data matched the configurations of relative limb phasing specified by each of the canonical gaits, we defined a metric of coherence based upon the Kuramoto model for networks of coupled oscillators (*Acebrón et al., 2005*). For each canonical gait, we defined a template of relative phases $\{\psi_k\}_{k=1}^6$. The template phases for each canonical gait are listed in *Table 2*. Then, given the set of instantaneous phases of the six limbs $\{\phi_k(t)\}_{k=1}^6$ we defined for each template pattern the global phase $\Phi(t)$ and coherence $r(t)$ by the relation

$$r(t)\exp\{i\Phi(t)\} = \tfrac{1}{6}\sum_{k=1}^6 \exp\{i(\phi_k(t)-\psi_k)\} \tag{5}$$

which corresponds exactly to the definition of the order parameters of the Kuramoto model with the addition of the phase template. The resulting coherence $r(t)$ ranges from a value of zero for asynchrony to a value of one for perfect alignment with the phase template.

## Dimensionality reduction with UMAP (*Figures 4*, *5* and *8*)

To generate low-dimensional representations of our data, we applied the nonlinear dimensionality reduction algorithm UMAP (*McInnes et al., 2018*). Using UMAP, we generated low-dimensional representations of $10^5$ randomly sampled segments of limb positional data, each with a half-window length of 100 ms. Given our sampling rate of 150 frames per second, each segment was thus an element of a 372-dimensional vector space. Before embedding, each time series was mean-subtracted on a per-segment basis, and each timepoint of each variable was standardized across trajectories to have zero mean and unit variance. This standardization does not substantially affect the resulting embedding, since all flies are roughly the same size, with the same mean limb positions. We note also that comparable results may be obtained with segments of length 100 ms or 400 ms rather than 200 ms (*Figure 4—figure supplement 4*). In *Figure 4B*, we colored the UMAP embedding by mean-subtracted limb position, a raw form of the input data.

## Synthetic canonical gait data (*Figure 3—figure supplement 1*; *Figure 4—figure supplement 5*)

To generate synthetic data matching canonical gait patterns, we adapted a model originally designed to generate quadruped gaits (*Righetti and Ijspeert, 2008*). Briefly, this model is composed of six coupled nonlinear oscillators, each with two degrees of freedom. We denote the state of the $i^{\text{th}}$ oscillator as $\{x_i, y_i\}$, where $x_i$ is the position of the limb in the direction parallel to the body axis, and $y_i$ is a variable which allows for the incorporation of feedback and defines whether the limb is in swing ($y_i<0$) or in stance phase ($y_i>0$). In this model, uneven duty ratios are implemented by introducing the state-dependent frequency

$$\omega_i = \omega_{\text{stane}} + \tfrac{\omega_{\text{swing}}-\omega_{\text{stane}}}{1+\exp(a\,y_i)} \tag{6}$$

where the stance frequency is given in terms of the swing frequency $\omega_{\text{swing}}$ and the duty factor $\beta$ as

$$\omega_{\text{stane}} = \tfrac{1-\beta}{\beta}\omega_{\text{swing}} \tag{7}$$

Then, defining $r_i^2 = x_i^2 + y_i^2$, the dynamical equations of the model are given as

$$\frac{\mathrm{d}x_i}{\mathrm{d}t} = \alpha(\mu - r_i^2)x_i - \omega_i y_i \tag{8}$$

$$\frac{\mathrm{d}y_i}{\mathrm{d}t} = \alpha(\mu - r_i^2)y_i + \omega_i x_i + \sum_{j=1}^6 k_{ij}y_j \tag{9}$$

where $\alpha$ode45 is a positive parameter governing the strength of self-interactions, $\mu$ode45 is a positive parameter governing the radius of the limit cycle, and $k_{ij}$ode45 is a coupling matrix. This system of nonlinear ordinary differential equations admits stable limit cycles for a broad range of coupling matrices (*Righetti and Ijspeert, 2008*). By adjusting the coupling matrix and the duty cycle appropriately, this model can produce synthetic limb traces with the duty cycle and phasing of the

four canonical gaits. We integrated these equations using Matlab's *ode45* medium-order adaptive timestep Runge-Kutta integrator. For each canonical gait we generated trajectories with swing durations iid from a uniform distribution corresponding to stepping frequencies between 5 and 12.5 Hz for a tripod gait. To correct for the fact that this model (with the coupling matrices chosen) is time-reversal symmetric, we adjust limb ordering post-hoc to enforce posterior-to-anterior propagation of metachronal waves. Finally, we add white Gaussian noise with a signal-to-noise ratio of 12.5 to the synthetic limb data to model the noise in our experimental data.

## Numerical modeling (*Figure 5*)

To build a generative model for phase dynamics during straight-walking, we make use of the fact that the cross-body phasing is constant, and first construct a simple rule-based discrete-time algorithm to generate metachronal waves along one side of the body. Following the empirical observation that the swing duration is approximately constant across all forward walking speeds, we fix the swing duration $\tau_{\text{swing}}$ and have as a free parameter the stance duration $\tau_{\text{stance}}$. As the swing and stance durations are dimensionful quantities, we define a timestep $\Delta t$ such that the ratios $\Delta t\ \tau_{\text{swing}}^{-1}$ and $\Delta t\ \tau_{\text{stance}}^{-1}$ are dimensionless. We denote the phases of the fore-, mid-, and hind-limbs at timestep $t$ as $\theta_1(t)$, $\theta_2(t)$, and $\theta_3(t)$, respectively. Defining phases modulo $2\pi$, we represent swings by values less than $\pi$, and stances by values greater than $\pi$. We fix a desired total number of timesteps $N$, and an initial condition $\{\theta_i(0)\}_{i=1}^3$.

Since the metachronal waves propagate from the posterior limb to the anterior limb, we model the phase dynamics of the posterior limb as being independent from those of other limbs; at each timestep its phase is simply incremented by $\Delta t\ \pi\ \tau_{\text{swing}}^{-1}$Algorithm 1 during swing and by $\Delta t\ \pi\ \tau_{\text{stance}}^{-1}$Algorithm 1 during stance. We now must choose a mechanism to ensure that the forelimb and hindlimb swing after their respective posterior neighbors. A simple means to do this is to reduce the rate of a given limb's phase advance during swing phase if its anterior neighbor is also in swing phase. We make the arbitrary but simple choice that the rate of phase advance is halved. These choices lead to the feedforward rule-based algorithm to generate the phase dynamics described in pseudocode in **Algorithm 1**. Conceptually, this is similar to classical rule-based algorithms for generating robot limb placement targets (*Wettergreen and Thorpe, 1992*; *Kwak and McGhee, 1989*; *Song and Waldron, 1987*).

---

**Algorithm 1** Rule-based generation of metachronal waves.

---

**Parameters** $\Delta t$, $\tau_{\text{swing}}$, $\tau_{\text{stance}}$, $N$, $\{\theta_i(0)\}_{i=1}^3$

**for** $t = 1, 2, 3, \ldots, N$ **do**

 **if** $\theta_3(t-1) < \pi\ mod\ 2\pi$ **then**

 $\theta_3 \leftarrow \theta_3(t-1) + \Delta t\ \pi\ \tau_{swing}^{-1}\ mod\ 2\pi$

 **else**

 $\theta_3 \leftarrow \theta_3(t-1) + \Delta t\ \pi\ \tau_{stance}^{-1}\ mod\ 2\pi$

 **end if**

 **if** $\theta_2(t-1) < \pi\ mod\ 2\pi$ **and** $\theta_3(t-1) < \pi\ mod\ 2\pi$ **then**

 $\theta_2 \leftarrow \theta_2(t-1) + 2^{-1}\Delta t\ \pi\ \tau_{swing}^{-1}\ mod\ 2\pi$

 **else if** $\theta_2(t-1) < \pi\ mod\ 2\pi$ **then**

 $\theta_2 \leftarrow \theta_2(t-1) + \Delta t\ \pi\ \tau_{swing}^{-1}\ mod\ 2\pi$

 **else**

 $\theta_2 \leftarrow \theta_2(t-1) + \Delta t\ \pi\ \tau_{stance}^{-1}\ mod\ 2\pi$

 **end if**

 **if** $\theta_1(t-1) < \pi\ mod\ 2\pi$ **and** $\theta_2(t-1) < \pi\ mod\ 2\pi$ **then**

 $\theta_1 \leftarrow \theta_1(t-1) + 2^{-1}\Delta t\ \pi\ \tau_{swing}^{-1}\ mod\ 2\pi$

 **else if** $\theta_2(t-1) < \pi\ mod\ 2\pi$ **then**

---

*Continued on next page*

---

$$\theta_1 \leftarrow \theta_1(t-1) + \Delta t\, \pi\, \tau_{swing}^{-1}\, mod\, 2\pi$$

else

$$\theta_1 \leftarrow \theta_1(t-1) + \Delta t\, \pi\, \tau_{stance}^{-1}\, mod\, 2\pi$$

end if

end for

We now note that **Algorithm 1** simply implements a forward Euler method for integrating the following system of ordinary differential equations:

$$\frac{d\theta_1}{dt} = \frac{\pi}{\tau_{\text{swing}}} \frac{f_{\text{swing}}(\theta_1)}{1 + f_{\text{swing}}(\theta_2)} + \frac{\pi}{\tau_{\text{stance}}} f_{\text{stance}}(\theta_1) \tag{10}$$

$$\frac{d\theta_2}{dt} = \frac{\pi}{\tau_{\text{swing}}} \frac{f_{\text{swing}}(\theta_2)}{1 + f_{\text{swing}}(\theta_3)} + \frac{\pi}{\tau_{\text{stance}}} f_{\text{stance}}(\theta_2)$$

$$\frac{d\theta_3}{dt} = \frac{\pi}{\tau_{\text{swing}}} f_{\text{swing}}(\theta_3) + \frac{\pi}{\tau_{\text{stance}}} f_{\text{stance}}(\theta_3)$$

where we have implemented the discrete-time rules using indicator functions for swing and stance phase, defined as

$$f_{\text{swing}}(\theta) = \begin{cases} 1 \ if\ \theta < \pi\ mod\ 2\pi \\ 0\ otherwise \end{cases} \tag{11}$$

and

$$f_{\text{stance}}(\theta) = \begin{cases} 1 \ if\ \theta > \pi\ mod\ 2\pi \\ 0\ otherwise \end{cases} \tag{12}$$

respectively. As shown in *Figure 5*, this simple model suffices to generate metachronal waves that appear qualitatively similar to those observed empirically. We note that this system of piecewise-linear ODEs could be replaced with a system that uses continuous, differentiable approximations for the indicator functions. However, this would introduce additional parameters governing the sharpness of the approximation. As the piecewise-continuous system is well-behaved but for the discontinuities, it may be numerically integrated with satisfactory accuracy using a fixed-step low-order Runge-Kutta method. Therefore, we chose to integrate the discontinuous rule-based model.

To generate six-legged phase dynamics, we coupled together two such generative models for metachronal waves. Since cross-body antiphase is conserved across forward walking speeds, our model is chiefly concerned with the generation of metachronal waves. The nature of the cross-body coupling is not central to our model; rather, it is a convenience used to generate phase dynamics for all the limbs in a feedforward manner. With this in mind, we chose to enforce cross-body antiphase by adding additional divisive terms depending on the sine of the contralateral phase differences. In doing so, we introduce an additional parameter $0 < \alpha < 1$ governing the strength of the cross-body coupling. In *Figure 5*, a value of $\alpha = \frac{1}{8}$ was used, though we find empirically that varying the value of $\alpha$ between $\frac{1}{8}$ and $\frac{1}{2}$ does not produce qualitative changes in the behavior of the model. Then, denoting phases of the left forelimb, left midlimb, left hindlimb, right forelimb, right midlimb, and right hindlimb as $\theta_1$ through $\theta_6$, respectively, the dynamical equations of the six-limb model are

$$\frac{d\theta_1}{dt} = \frac{\pi}{\tau_{\text{swing}}}\frac{f_{\text{swing}}(\theta_1)}{1+f_{\text{swing}}(\theta_2)+\alpha\sin(\theta_4-\theta_1)} + \frac{\pi}{\tau_{\text{stance}}}f_{\text{stance}}(\theta_1)$$

$$\frac{d\theta_2}{dt} = \frac{\pi}{\tau_{\text{swing}}}\frac{f_{\text{swing}}(\theta_2)}{1+f_{\text{swing}}(\theta_3)+\alpha\sin(\theta_5-\theta_2)} + \frac{\pi}{\tau_{\text{stance}}}f_{\text{stance}}(\theta_2)$$

$$\frac{d\theta_3}{dt} = \frac{\pi}{\tau_{\text{swing}}}\frac{f_{\text{swing}}(\theta_3)}{1+\alpha\sin(\theta_6-\theta_3)} + \frac{\pi}{\tau_{\text{stance}}}f_{\text{stance}}(\theta_3)$$

$$\frac{d\theta_4}{dt} = \frac{\pi}{\tau_{\text{swing}}}\frac{f_{\text{swing}}(\theta_4)}{1+f_{\text{swing}}(\theta_5)+\alpha\sin(\theta_1-\theta_4)} + \frac{\pi}{\tau_{\text{stance}}}f_{\text{stance}}(\theta_4)$$

$$\frac{d\theta_5}{dt} = \frac{\pi}{\tau_{\text{swing}}}\frac{f_{\text{swing}}(\theta_5)}{1+f_{\text{swing}}(\theta_6)+\alpha\sin(\theta_2-\theta_5)} + \frac{\pi}{\tau_{\text{stance}}}f_{\text{stance}}(\theta_5)$$

$$\frac{d\theta_6}{dt} = \frac{\pi}{\tau_{\text{swing}}}\frac{f_{\text{swing}}(\theta_6)}{1+\alpha\sin(\theta_3-\theta_6)} + \frac{\pi}{\tau_{\text{stance}}}f_{\text{stance}}(\theta_6)$$

(13)

We integrated this system of ODEs with the parameters given in *Table 3* using Heun's method, a second-order explicit Runge-Kutta integrator (*Ascher and Petzold, 1998*) with a timestep of 0.025 ms, as it provides a reasonable balance of computational efficiency and numerical stability.

## Optogenetic manipulations (*Figure 8*)

For optogenetic experiments, we augmented the arena used in free-walking experiments by mounting a DLP Lightcrafter 4500 projector (Texas Instruments, Dallas, TX) below the walking surface. During experiments, flies were loaded into the arena and allowed to acclimate for 20 min prior to recording. For each experiment, 12–15 females were allowed to walk freely within the arena and were recorded from below for 1.1 hr. Throughout the recording, at 4 s intervals, 8 ms full-field flashes of ~ 0.05 mW/mm$^2$ red light (peak 624 nm) were projected into the arena in order to optogenetically activate moonwalker neurons of the flies. In both experimental and random trigger control conditions, we selected trajectories where the fly was walking > 5 mm/s on average in the 100 ms prior to stimulation. Additionally, in order to exclude stopping flies, we removed trajectories in which the flies' forward velocity was less than 0.1 mm/s at any point post stimulus. Data from seven separate recordings, with a total of 96 flies, were merged to generate the aggregate dataset analyzed in this manuscript.

## Visual stimuli (*Figure 8*)

Visual stimuli were designed using Matlab and its associated Psychophysics Toolbox (*Brainard, 1997*; *Kleiner et al., 2007*; *Pelli, 1997*). They were projected onto a virtual flat torus masked with a disk placed on a flat screen 29.5 mm above the free-walking arena using a DLP Lightcrafter projector (Texas Instruments, Dallas, TX). The spatial resolution of the screens was ~ 0.3° and the screen update rate was 180 Hz. All visual stimuli used green light (peak 520 nm) with mean luminance of 100 cd/m$^2$. The video data were aligned to the stimulus by presenting periodic flashes measured with a photodiode. All stimuli were presented in 50 ms bouts separated by 950 ms interleaves. The stimulus presented consisted of an array of 15°-diameter white and black circular dots, placed on a mean-grey background at locations iid from a uniform distribution with a density of 10%. Overlap between dots was handled such that the contrast of the stimulus did not increase (two white dots add to white, two black dots sum to black, and a white and black dot sum to gray). The resulting image was translated rigidly in a randomly selected cardinal direction during stimulation, and was static during the interleave. Three velocities of translation, 720, 960, and 1440 °/s, were grouped and used as the visual stimulus condition in *Figure 8*. Trajectories were selected in a manner identical to the optogenetic experiments described above. For these experiments, the flies were backlit with infrared light (peak 850 nm) at an intensity of ~ 2 µW/mm$^2$. Data from three separate recordings, with a total of 44 flies, were merged to generate the aggregate dataset analyzed in this manuscript.

**Table 3.** Parameters used for model in *Figure 5*.

| Parameter | Value |
| --- | --- |
| $\alpha$ | 1/8 |
| $\tau_{\text{swing}}$ | 40 ms |
| $\tau_{\text{stance}}$ | 40–210 ms |

**Table 4.** Two-sample Monte Carlo resampling tests using the Kuiper V-statistic against the null hypothesis that the distribution of phases at yaw extrema is indistinguishable from that over all instants ($10^5$ permutations, N = 8 videos).

| Limb | V-statistic | p-Value | 95% CI for p-value |
|---|---|---|---|
| O1 | $1.08 \times 10^{-1}$ | $<10^{-5}$ | $[0, 3.68 \times 10^{-5}]$ |
| O2 | $1.67 \times 10^{-1}$ | $<10^{-5}$ | $[0, 3.68 \times 10^{-5}]$ |
| O3 | $1.77 \times 10^{-1}$ | $<10^{-5}$ | $[0, 3.68 \times 10^{-5}]$ |
| I1 | $1.07 \times 10^{-1}$ | $<10^{-5}$ | $[0, 3.68 \times 10^{-5}]$ |
| I2 | $2.33 \times 10^{-1}$ | $<10^{-5}$ | $[0, 3.68 \times 10^{-5}]$ |
| I3 | $1.16 \times 10^{-1}$ | $<10^{-5}$ | $[0, 3.68 \times 10^{-5}]$ |

## Statistics

All statistical analysis was conducted using Matlab 9.4 (The MathWorks, Natick, MA). Throughout this manuscript, presented error bars are bootstrapped 95% confidence intervals obtained using the bias-corrected and accelerated percentile method (*Efron, 1987*) unless otherwise indicated. In *Figure 1*, error bars for both body and limb variables were generated from bootstraps over videos (eight videos total from eight experiments, each with approximately 1/8 of the total data). In *Figures 3*, *6* and *7*, error bars for analyses of relative limb phases and coherences were also generated by bootstrapping over videos. In *Figure 8*, unlike other analyses, the confidence intervals were generated by bootstrapping over individual selected fly walking trajectories. In *Figure 8*, distributions of stance durations were compared using a two-sample Kolmogorov-Smirnov test. Throughout this manuscript, circular probability density functions of limb relative phases were estimated using kernel methods (*Batschelet, 1981*; *Fisher, 1989*). In *Figure 7*, to test against the null hypothesis that the distribution of phases at yaw extrema is indistinguishable from that over all instants, we performed two-sample Monte Carlo resampling tests (*Dwass, 1957*; *Gandy, 2009*) using the Kuiper V-statistic (*Batschelet, 1981*; *Kuiper, 1960*). We used $10^5$ permutations, and computed Clopper-Pearson confidence intervals for the resulting p-values (*Clopper and Pearson, 1934*; *Dwass, 1957*; *Gandy, 2009*) (*Table 4*). As the value of the test statistic for the null distribution did not exceed the observed test statistic in $10^5$ permutations, this test only allows us to bound the true p-value from above.

## Data and code availability

The datasets analyzed in this work are available from the Dryad Digital Repository: https://doi.org/10.5061/dryad.3p9h20r. Matlab code to integrate all models, reproduce all statistical analyses, and generate all figure panels is available at https://github.com/ClarkLabCode/GaitPaperCode (*DeAngelis et al., 2019*; copy archived at https://github.com/elifesciences-publications/GaitPaperCode).

## Acknowledgements

This work benefitted from many helpful conversations with M Venkadesan, as well as analysis by U Piterbarg. BDD was supported by an NSF GRF and a Gruber Science Fellowship. DAC and this project were supported by the Smith Family Foundation, NIH R01EY026555, a Searle Scholar Award, and a Sloan Fellowship in Neuroscience.

## Additional information

### Funding

| Funder | Grant reference number | Author |
|---|---|---|
| National Institutes of Health | R01 EY026555 | Brian DeAngelis<br>Damon A Clark |

| | | |
|---|---|---|
| National Science Foundation | GRFP | Brian DeAngelis |
| Richard and Susan Smith Family Foundation | | Brian DeAngelis<br>Damon A Clark |
| Alfred P. Sloan Foundation | | Damon A Clark |
| Kinship Foundation | Searle Scholar Award | Brian DeAngelis<br>Damon A Clark |

The funders had no role in study design, data collection and interpretation, or the decision to submit the work for publication.

#### Author contributions
Brian D DeAngelis, Jacob A Zavatone-Veth, Conceptualization, Data curation, Software, Formal analysis, Investigation, Visualization, Methodology, Writing—original draft, Writing—review and editing; Damon A Clark, Conceptualization, Supervision, Funding acquisition, Investigation, Writing—original draft, Project administration, Writing—review and editing

#### Author ORCIDs
Brian D DeAngelis (iD) https://orcid.org/0000-0001-9418-7619
Jacob A Zavatone-Veth (iD) https://orcid.org/0000-0002-4060-1738
Damon A Clark (iD) https://orcid.org/0000-0001-8487-700X

#### Decision letter and Author response
Decision letter https://doi.org/10.7554/eLife.46409.sa1
Author response https://doi.org/10.7554/eLife.46409.sa2

## Additional files

#### Supplementary files
• Transparent reporting form

#### Data availability
The datasets analyzed in this work are available from the Dryad Digital Repository: https://doi.org/10.5061/dryad.3p9h20r. Matlab code to reproduce all statistical analyses and figure panels is available at https://github.com/ClarkLabCode/GaitPaperCode (copy archived at https://github.com/elifesciences-publications/GaitPaperCode).

The following dataset was generated:

| Author(s) | Year | Dataset title | Dataset URL | Database and Identifier |
|---|---|---|---|---|
| DeAngelis B, Zavatone-Veth JA, Clark DA | 2019 | Data from: The manifold structure of limb coordination in walking *Drosophila* | http://dx.doi.org/10.5061/dryad.3p9h20r | Dryad Digital Repository, 10.5061/dryad.3p9h20r |

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
