## [Decision Letter]

Thank you for submitting your article "The manifold structure of limb coordination in walking *Drosophila*" for consideration by *eLife*. Your article has been reviewed by 3 peer reviewers, and the evaluation has been overseen by Ronald Calabrese as the Senior and Reviewing Editor. The reviewers have opted to remain anonymous.

The reviewers have discussed the reviews with one another and the Reviewing Editor has drafted this decision to help you prepare a revised submission.

Summary:

This work investigates the repertoire of forward locomotion dynamics in the fruit fly (*D. melanogaster*) using a combination of limb tip and centroid tracking, dimensionality reduction and mathematical modeling, and neuronal perturbations. The key claims are that the speed of forward locomotion are controlled using a single control parameter, and that the traditional division of walking patterns into alternating tripod, tetrapod, and wave gaits is not apparent in spontaneous walking and is, accordingly, better understood through the manifold framework proposed here. Lastly, the authors perform both optogenetic and sensory perturbations during spontaneous locomotion, finding that the flies altered their movements along the trajectory of the manifold described.

Essential revisions:

There is much to interest the fly community and the greater neuroscience community in the thorough analysis presented and the interesting manifold structure described.

Several concerns detailed in the expert reviews should be addressed.

1) The UMAP embedding (dimensionality reduction) is well-suited for preserving topological features of a data set requiring warping of global metrics in order to preserve local topological structure. PCA may do a better job of describing the full dynamic range of the variables than the UMAP does. The authors should directly compare PCA to the UMAP manifold and should be very clear about what features of the data UMAP is preserving.

2) The effect of error on gait definitions (e.g. those in Figure 3) is not explored/discussed. The authors make stance/swing definitions based on the smoothed instantaneous velocity of the limb (with the tracking error threshold chosen as the separation threshold). While this is a sensible approach given the type data obtained here there is no analysis or discussion about how error in this value might affect the results found.

3) Previous studies have shown that limb coordination patterns in *Drosophila* are highly variable and that while there are changes with speed that the overall shifts are more continuous than observed in many other animals. The authors would be better served by not emphasizing how this work shows that flies do not exhibit discrete gaits and emphasizing more the structure of the manifold discovered and how it is relevant to locomotion.

4) The paper would be substantially improved if the coupled phase oscillator model were motivated and presented more clearly.

5) A common feature in the gait transition literature is hysteresis – the transition occurs at a different speed going from slow-to fast than fast-to-slow. Either adapting the analysis to include this potential effect (or explaining why it is not relevant here) is a necessary addition to the manuscript.

6) The spontaneous turning data from your other *eLife* submission and its relationship to the manifold described would enhance this paper and further take the emphasis off the current discrete gaits controversy. This is not a requirement but a suggestion.

*Reviewer #1:*

Note: I am also a reviewer of your other *eLife* submission, so several of my comments will mirror those for that paper, due to the similarity of the methods used.

The manuscript by DeAngelis and colleagues investigates the repertoire of forward locomotion dynamics in the fruit fly (*D. melanogaster*) using a combination of limb tip and centroid tracking, dimensionality reduction, mathematical modeling, and neuronal perturbations. The key claims are that the speed of forward locomotion are controlled using a single control parameter, and that the traditional division of walking patterns into alternating tripod, tetrapod, and wave gaits is not apparent in spontaneous walking (and is, accordingly, better understood through the manifold framework proposed here). Lastly, the authors perform both optogenetic and sensory perturbations during spontaneous locomotion, finding that the flies altered their movements along the trajectory of the found manifold.

Overall, I thought that this manuscript was creatively performed and would be of general interest to the community. I do have some serious reservations about the findings, however, and would like to see some further analysis of the data before recommending publication. Specifically, I have questions about the authors' choice of embedding to find the manifold, the effects of footfall error on their results and interpretation of their results in light of gait hysteresis. If these concerns are resolved satisfactorily, however, I envision that this paper will be well-suited to publication in *eLife*.

1) After reading and digesting the paper (as well as the co-submitted paper), I remain a bit perplexed why the authors decided to use UMAP for the dimensionality reduction technique upon which many of their results lie. UMAP is well-suited for preserving topological features of a data set requiring warping of global metrics in order to preserve local topological structure. Despite this, the authors only use UMAP as a global coordinate, treating it like one often observed PCA (which does the opposite, preserving global metrics while warping local ones), pointing at directions where yaw or lateral or forward velocities alter. In fact, the results from Figure 2—figure supplement 1 in Zavatone-Veth et al. show that PCA actually does a better job of describing the full dynamic range of the variables than the UMAP does. As a result, it also would be interesting to see what PCA provides for the kinematic variables.

I don't make these points as merely a technicality, however. As many of the results in the paper come from this embedding, it is important to actually note what the structure of this manifold is. Moreover, the density within the manifold is not shown in any of the plots as well. Are there islands of high density with valleys in between? Or is It smoothly distributed? I think that more care is needed to more fully-characterize the manifold that is the one of the key mathematical descriptions in this manuscript (and is in fact in the title!).

2) Perhaps my largest concern with the paper, however, is the effect of error on gait definitions (e.g. those in Figure 3). The authors make stance/swing definitions based on the smoothed instantaneous velocity of the limb (with the tracking error threshold chosen as the separation threshold). While this is a sensible approach given the type data that they obtain here, as far as I saw in this manuscript at any rate, I did not see any discussion about how error in this value might affect the found results. This is rather important, as many of the disagreements with the previous literature about the discreteness of individual gaits rely on these measurements. I think that it would be important to see how the results change if the threshold for swing/stance was changed within some reasonable limits, or if swing/stance was assigned using a confidence (using, say, a logistic function of the limb speed). Specifically, I would be curious to see if the 2-wave vs. 3-wave and 6-wave results could be altered by changing this threshold.

That being said, the values for seem *that* far off. The results from Mendes et al., 2013, for example, are built upon a much more accurate measurement of swing/stance determination, since the footfalls are directly measured using frustrated total internal reflection (although the authors should please correct me if they feel this assessment is incorrect). The results in Figure 1 here are similar to those found in Figure 2 of that paper, so no major tweaking is likely needed, but some assessment of the effect of potential error is important. Relatedly, it might be useful to see what happens to the "tripod index" as defined in the Medes paper with the data collected here. Are the results replicated?

3) Related to both the previous comments, I think that in some senses, this paper is offering a potentially-false dichotomy between discrete and continuous gaits. For example, although the fly may have some preferred forward velocities or gaits that are peaks in the distribution, there may be non-zero valleys between them that, although rare, are important parts of the overall manifold that describes locomotion. For example, in Figure 4—figure supplement 1, the separate regions are potentially the result of the fact that there is *no* intermediate data. Because UMAP explicitly exaggerates topological differences, my guess is that the inclusion of a relatively small number of intermediate gaits will link the discrete rings together into something that looks more like the real data. I also worry that the phase error potentially-introduced from the uncertainty of the stance/swing transition time might affect the phase difference peaks in Figure 3 and Figure 3—figure supplements 1 and 2.

4) A common feature in the gait transition literature is hysteresis – the transition occurs at a different speed going from slow-to fast than fast-to-slow. All of the analyses here fundamentally assume that gait is a single-valued function of speed, however, which may cause for some misinterpretations of the data, since all spread of the distribution would just be assigned to a single mean value. Either adapting the analysis to include this potential effect (or explaining why it is not relevant here) is a necessary addition to the manuscript.

Reviewer #2:

In this paper, DeAngelis and colleagues investigate the temporal organization of the fly walking gait. They use a simple tracking algorithm to measure the body orientation and distal leg position of flies walking in a flat, featureless environment. They confirm previous observations that stance duration changes as a function of walking speed. They then extracted swing and stance phases from each leg and used these patterns to classify different walking gaits. They find that flies typically have three feet in stance (tripod gait), but they observe that flies occasionally (and transiently) have 4 or 5 feet in stance, particularly at low walking speeds. Contralateral limbs reliably swing in antiphase, while ipsilateral limb coordination vary a bit more. A low-dimensional embedding visualization of limb coordinates resulted in cyclic (global oscillator phase) and one linear coordinate (stepping frequency). Overall, this suite of analyses suggest that there is no clear differentiation of walking behavior into distinct gaits (e.g., tripod vs. tetrapod). The authors then construct a model that can predict the continuum of walking behavior by tuning of a single parameter. Finally, they find that perturbing walking behavior through activation of the Moonwalker Descending Neuron or visual stimulation alters stance duration, and thus induces a shift along the length of the walking manifold.

Overall, the paper is clearly organized and well-written. Although the subject of the fly walking gait is already well-trodden, this is certainly the most exhaustive and definitive treatment to date. I think it is a very strong paper and have only a handful of suggestions to improve it prior to publication.

1) My biggest point of confusion was the coupled phase oscillator model. The paper would be improved if this model were motivated and presented more clearly. The equations are not intuitive at first glance (e.g., the 1/sin terms, the discontinuous indicator functions); is there an easy interpretation of them? And does the specific form of the model matter? How is it related to the other coupled oscillator models in the literature? I would have liked to see a sensitivity analysis that provides some intuition for how the model works and which parameters matter. For example, τ_stance_ is the relevant parameter that the authors tune to get different inter-limb coordination patterns, but this is not clearly explained (especially in the main text). Finally, how does this model contribute to our understanding or intuition of the control strategy used by the fly to coordinate walking? The authors should include a deeper discussion of the motivation for and interpretation of this model.

Reviewer #3:

This paper is notable for collecting a very comprehensive dataset of kinematic gait patterns in *Drosophila* over a wide range of spontaneous and evoked speed changes. While the optogenetic and visually induced changes are nice as supporting evidence, the core of the paper comes from a detailed and thorough gait analysis. As such the paper has some very good insights when connected to modern dimensionality reduction and some simple models of oscillations, but needs to distinguish itself carefully from the long history of papers that also do gait analysis especially in *Drosophila*.

1) The structure of the argument is that animals in general are thought to have distinct gaits. The authors cite a number of studies on larger legged animals for this, but in those cases walking is an out of phase oscillation of gravitational potential energy and kinetic energy (the inverted pendulum model) and running is in-phase oscillations (the spring-loaded inverted pendulum). The transition in gaits in many animals from insects to horses corresponds to the transition between these underlying dynamics If this transition is unlikely (although possible?) for *Drosophila* given their size, it is not clear why we would expect distinct gaits even if larger animals have them.

More critical to the conclusions drawn about *Drosophila*, previous studies (Wosnitza et al., 2012, and Mendes et al., 2013) My reading of these previous studies is that they show that limb coordination patterns in *Drosophila* are highly variable and that while there were changes with speed that the overall shifts are more continuous than observed in many other animals. These are cited in the paper but the relationship of their conclusions to this paper are only discussed superficially. For example Wosnitza et al., 2012 conclude that their findings "imply that *Drosophila*'s walking behavior is more flexible than previously thought (Strauss and Heisenberg, 1990): there are no clearly separable gaits and, more specifically, the neural controller producing inter-leg coordination is not restricted to a fixed tripod pattern." This is very close to the initial conclusions of this paper.

So while some data from those studies do fit tetrapod and tripod gaits (as do some in this study), the conclusion that *Drosophila* use a variety kinematic patterns that encompass the canonical gaits is already somewhat established. Therefore the hypothesis of distinct gaits in this paper seems like a bit of a strawman. The authors do make some nice analyses of these data and show the point perhaps more convincingly than in previous work. However, prior to the manifold section, the initial presentation of the data did not leave me thinking that I had learned something new. I recognize that the authors may not agree with interpretations in previous works, but the onus is on the authors to clearly articulate what cannot be concluded from the previous work that their work in turn justifies.

More importantly this whole argument of distinct gaits takes away from what for me was the real impact of the paper. The central advance of the paper is the finding that kinematics patterns across a large range of speeds can collapse down onto a manifold with a single parameter governing speed variation. The authors are still making an advance here, but the motivation of the introduction and the conclusion of the first sentence of the Discussion highlight the continuum vs. discrete nature of *Drosophila* gaits. The emphasis should be about the manifold.

2) My concern about the manifold discussion is how this approach is different from the many other dimensionality reductions that have been done on gait. What really sets it apart from the t-SNE analyses of behavior that show all walking lying in a low dimensional cluster (Berman et al., 2014) and the reduced phase and oscillator models of stick insects and cockroaches (e.g. Couzin-Fuchs et al., 2015). I think there is novelty here but it must be articulated more in context.

---

## [Author Response]

Essential revisions:There is much to interest the fly community and the greater neuroscience community in the thorough analysis presented and the interesting manifold structure described.Several concerns detailed in the expert reviews should be addressed.1) The UMAP embedding (dimensionality reduction) is well-suited for preserving topological features of a data set requiring warping of global metrics in order to preserve local topological structure. PCA may do a better job of describing the full dynamic range of the variables than the UMAP does. The authors should directly compare PCA to the UMAP manifold and should be very clear about what features of the data UMAP is preserving.

Thank you for this suggestion. We initially considered using PCA to visualize our locomotion dataset. Unfortunately, PCA does not provide a particularly clarifying description of the overall structure of variability in limb movement patterns. We have added a supplementary figure (Figure 4—figure supplement 1) to illustrate this point. The principal components of our limb kinematic data occur in degenerate, approximately phase-shifted pairs. Therefore, though the projection of these data into the first two principal components provides some information about their phasic structure, PCA does not allow for easy visualization of the full structure of variability in two or three dimensions. Specifically, the relationship between the projection and the frequency of stepping is multivalued.

We would like to highlight that the principal component analysis presented in the companion manuscript was performed on the body kinematic variables. Due to long-timescale correlations among those coordinates, the body kinematic data are approximately three-dimensional, even when segments of trajectories are embedded. Therefore, projection into two dimensions using PCA yields more informative results than the application of PCA to the higher-dimensional limb data. We have adjusted the prose to address the difficulties encountered with PCA and t-SNE, another commonly used dimensionality-reduction technique.

We acknowledge that UMAP is a relatively new dimensionality reduction technique. To further clarify the features of the data that are preserved by UMAP, we have added a supplementary figure (Figure 4—figure supplement 2) including the additional visualizations requested by Reviewer #1. We have also expanded the prose to more clearly describe the optimization performed by the UMAP algorithm. We have added citations to several other publications, notably (Becht et al., 2019), in which it was applied to reveal the global topology of high-dimensional gene expression data, consistent with its usage in this manuscript.

2) The effect of error on gait definitions (e.g. those in Figure 3) is not explored/discussed. The authors make stance/swing definitions based on the smoothed instantaneous velocity of the limb (with the tracking error threshold chosen as the separation threshold). While this is a sensible approach given the type data obtained here there is no analysis or discussion about how error in this value might affect the results found.

Thank you for this suggestion. We have taken two steps to address the sensitivity of the swing/stance analyses. First, the analyses presented in Figure 3 are independent of our determination of swing and stance, as the instantaneous phase measurements used throughout the figure are estimated directly from raw limb position time series. We have modified the prose to state this more clearly. Second, in Figure 2, the analyses presented do depend on swing/stance determinations. If one believed that measurement noise strongly affected these analyses, one would expect that lowering the separation threshold would accentuate the effect. To address this concern, we replicated Figure 2 using a separation threshold of 10 pixels per frame rather than 20 pixels per frame as in the main figure. As shown in Author response image 1, the two-cycle in the instantaneous number of feet in stance is preserved even under halving of the separation threshold. Thus, we believe this analysis is relatively insensitive to measurement noise in our swing-stance definition. We have updated the prose in the manuscript to describe this sensitivity analysis.

**Author response image 1. respfig1:** Replication of Figure 2 with halved swing/stance separation threshold.

Following this comment, we also wanted to test whether our phase-based analyses were sensitive to noise in the limb measurements. To test the noise sensitivity of the discrete-time analytic signal method we used for phase estimation, we simulated a set of trajectories using our canonical gait model. As this phase estimation method is known to perform best with waveforms with duty cycle equal to 1/2 (Boashash and Reilly, 1992; Lamb and Stöckl, 2014), we chose to simulate canonical right tetrapod gaits, which have a duty cycle equal to 2/3, so that we tested the algorithm where it might not work as well. We then corrupted the resulting simulated limb position data with additive white Gaussian noise to yield signal-to-noise ratios (std of signal/std of noise) of 12.5 (matching Figure 3—figure supplement 1 and Figure 4—figure supplement 4; also roughly matching our estimate of the experimental signal to noise) and 1.25. Following the protocol applied to our experimental data, we then estimated the phase from these noisy traces and smoothed the resulting estimate using a third-order Savitzky-Golay filter. As shown in Author response image 2, the spread of the relative phase distribution increased with decreasing SNR, but the location and number of modes were not qualitatively altered. Therefore, our phase estimation method appears to be relatively robust to noise.

**Author response image 2. respfig2:** Analysis of phase extraction with high SNR (A and B) and low SNR (C and D).

3) Previous studies have shown that limb coordination patterns in *Drosophila* are highly variable and that while there are changes with speed that the overall shifts are more continuous than observed in many other animals. The authors would be better served by not emphasizing how this work shows that flies do not exhibit discrete gaits and emphasizing more the structure of the manifold discovered and how it is relevant to locomotion.

We agree that previous studies have presented evidence that limb coordination patterns in *Drosophila* are highly variable, and believe that the primary contribution of our study is to characterize the structure of that variability. We have adjusted the prose throughout the manuscript to better reflect this understanding. In the literature, the structure of variability in walking patterns in many animals has most often been studied using the framework of distinct gaits, so we believe it is important to address the question of gaits. Indeed, this framework has persisted in the descriptions (Aminzare and Holmes, 2018; Aminzare et al., 2018; Mendes et al., 2013; Pereira et al., 2018; Strauss and Heisenberg, 1990) and analyses (Berman et al., 2014; Mendes et al., 2013; Pereira et al., 2018) used in the *Drosophila* locomotion literature, despite the observations of variable coordination patterns. We have therefore retained some analysis of and discussion of gaits, while simultaneously shifting the primary focus of the manuscript to describing the variability in coordination patterns. Adding results about spontaneous turning (Essential Revision 6) has also decreased the focus on gaits.

4) The paper would be substantially improved if the coupled phase oscillator model were motivated and presented more clearly.

Thank you for this suggestion. We have updated the main text and Materials and methods to clarify and motivate the description of the coupled phase oscillator model.

We have updated the prose to clarify the rule-based nature of the model, and its relation to conceptually similar rule-based algorithms for robot limb placement, such as (Kwak and McGhee, 1989; Song and Waldron, 1987; Wettergreen and Thorpe, 1992). To that end, we now present an algorithmic description in pseudocode. Additionally, we regret the lack of clarity surrounding the various model parameters. Parameters other than the stance duration are not critical to the model; we have updated the prose to clarify this. We therefore did not include analyses of the sensitivity to parameters other than the stance duration.

We have also updated the prose to better highlight the intuition generated by the model regarding the fly’s walking control strategy. Briefly, in this framework a single descending command conveying the desired stance duration would suffice to control the forward speed of the fly. This single control parameter permits a simpler control architecture than models in which inter-limb coupling is actively varied with walking speed. On each side of the fly, a posterior-to-anterior set of limb swings can be thought of as a single event. The two sides of the body are then coupled to each other in a manner that maintains a half-cycle difference in phase between contralateral limbs.

5) A common feature in the gait transition literature is hysteresis – the transition occurs at a different speed going from slow-to fast than fast-to-slow. Either adapting the analysis to include this potential effect (or explaining why it is not relevant here) is a necessary addition to the manuscript.

We agree that, based upon the literature, it is important to address the possibility of hysteresis in our experimental data. As we do not observe any evidence of multiple fixed points in our limb coordination data, it is difficult to directly address the question of hysteresis. To address the possibility that multiple preferred limb coordination patterns exist at certain speeds, we have added a supplementary figure (Figure 3—figure supplement 2) showing the distributions of pairwise relative phases conditioned with high resolution on forward velocity. At all forward walking speeds, these distributions are unimodal, which leads us to believe that our dataset does not include substantial hysteretic effects.

6) The spontaneous turning data from your other eLife submission and its relationship to the manifold described would enhance this paper and further take the emphasis off the current discrete gaits controversy. This is not a requirement but a suggestion.

Thank you for this suggestion. We agree that incorporating the spontaneous turning data originally presented in our rejected submission would enhance this paper, and have done so.

Reviewer #1:Note: I am also a reviewer of your other eLife submission, so several of my comments will mirror those for that paper, due to the similarity of the methods used.The manuscript by DeAngelis and colleagues investigates the repertoire of forward locomotion dynamics in the fruit fly (*D. melanogaster*) using a combination of limb tip and centroid tracking, dimensionality reduction, mathematical modeling, and neuronal perturbations. The key claims are that the speed of forward locomotion are controlled using a single control parameter, and that the traditional division of walking patterns into alternating tripod, tetrapod, and wave gaits is not apparent in spontaneous walking (and is, accordingly, better understood through the manifold framework proposed here). Lastly, the authors perform both optogenetic and sensory perturbations during spontaneous locomotion, finding that the flies altered their movements along the trajectory of the found manifold.Overall, I thought that this manuscript was creatively performed and would be of general interest to the community. I do have some serious reservations about the findings, however, and would like to see some further analysis of the data before recommending publication. Specifically, I have questions about the authors' choice of embedding to find the manifold, the effects of footfall error on their results and interpretation of their results in light of gait hysteresis. If these concerns are resolved satisfactorily, however, I envision that this paper will be well-suited to publication in eLife.1) After reading and digesting the paper (as well as the co-submitted paper), I remain a bit perplexed why the authors decided to use UMAP for the dimensionality reduction technique upon which many of their results lie. UMAP is well-suited for preserving topological features of a data set requiring warping of global metrics in order to preserve local topological structure. Despite this, the authors only use UMAP as a global coordinate, treating it like one often observed PCA (which does the opposite, preserving global metrics while warping local ones), pointing at directions where yaw or lateral or forward velocities alter. In fact, the results from Figure 2—figure supplement 1 in Zavatone-Veth et al. show that PCA actually does a better job of describing the full dynamic range of the variables than the UMAP does. As a result, it also would be interesting to see what PCA provides for the kinematic variables.I don't make these points as merely a technicality, however. As many of the results in the paper come from this embedding, it is important to actually note what the structure of this manifold is. Moreover, the density within the manifold is not shown in any of the plots as well. Are there islands of high density with valleys in between? Or is It smoothly distributed? I think that more care is needed to more fully-characterize the manifold that is the one of the key mathematical descriptions in this manuscript (and is in fact in the title!).

Our primary response to this comment is under Essential Revision 1. To address the concerns regarding the specific structure of the manifold, we have added a supplementary figure (Figure 4—figure supplement 2) showing the parameterization of the manifold with cylindrical coordinates. Using this projection, we visualized the density of points within the manifold. The distribution along the phase dimension is roughly uniform. As one would expect given the nonuniform distribution of forward speeds, the density along the frequency dimension of the manifold is not uniform, though it changes smoothly. The projection into cylindrical coordinates also allows us to generate more easily interpretable visualizations relating the UMAP dimensions to limb phases and frequencies, as well as a clear visualization of how the radius of the manifold is related to the axial dimension.

2) Perhaps my largest concern with the paper, however, is the effect of error on gait definitions (e.g. those in Figure 3). The authors make stance/swing definitions based on the smoothed instantaneous velocity of the limb (with the tracking error threshold chosen as the separation threshold). While this is a sensible approach given the type data that they obtain here, as far as I saw in this manuscript at any rate, I did not see any discussion about how error in this value might affect the found results. This is rather important, as many of the disagreements with the previous literature about the discreteness of individual gaits rely on these measurements. I think that it would be important to see how the results change if the threshold for swing/stance was changed within some reasonable limits, or if swing/stance was assigned using a confidence (using, say, a logistic function of the limb speed). Specifically, I would be curious to see if the 2-wave vs. 3-wave and 6-wave results could be altered by changing this threshold.That being said, the values for seem that far off. The results from Mendes et al., 2013, for example, are built upon a much more accurate measurement of swing/stance determination, since the footfalls are directly measured using frustrated total internal reflection (although the authors should please correct me if they feel this assessment is incorrect). The results in Figure 1 here are similar to those found in Figure 2 of that paper, so no major tweaking is likely needed, but some assessment of the effect of potential error is important. Relatedly, it might be useful to see what happens to the "tripod index" as defined in the Medes paper with the data collected here. Are the results replicated?

Our primary response is in the comments to Essential Revision 2. Specifically, the results regarding the 2-, 3-, and 6- wave results were preserved under halving of the separation threshold, suggesting that choice of threshold does not strongly influence this finding. Additionally, we have reproduced in Author response image 3 the gait index analysis from (Mendes et al., 2013) on the individual example trajectories presented in Figure 3F-G; that analysis is shown in Author response image 3. Consistent with Mendes et al., 2013, we are capable of extracting individual trajectories with tripod-like and tetrapod-like characteristics under this measure. This, in combination with other presented analyses (Figures 1-2, specifically Figure 2C-D), suggests that our experimental data is largely consistent with data collected in previous studies of *Drosophila* walking. Importantly, the tetrapod-like patterns identified by this index are consistent with our smoothly varying generative model.

**Author response image 3. respfig3:** Example trajectories with gait index analysis from Mendes et al., 2013.

3) Related to both the previous comments, I think that in some senses, this paper is offering a potentially-false dichotomy between discrete and continuous gaits. For example, although the fly may have some preferred forward velocities or gaits that are peaks in the distribution, there may be non-zero valleys between them that, although rare, are important parts of the overall manifold that describes locomotion. For example, in Figure 4—figure supplement 1, the separate regions are potentially the result of the fact that there is no intermediate data. Because UMAP explicitly exaggerates topological differences, my guess is that the inclusion of a relatively small number of intermediate gaits will link the discrete rings together into something that looks more like the real data. I also worry that the phase error potentially-introduced from the uncertainty of the stance/swing transition time might affect the phase difference peaks in Figure 3 and Figure 3—figure supplements 1 and 2.

Our primary response is in the comments to Essential Revision 3. Regarding the specific analysis presented in Figure 4—figure supplement 1, we would like to note that we do not view the UMAP embedding as a direct line of evidence for the absence of distinct gaits. Rather, it is an ancillary line of evidence that is consistent with the existence of the continuum suggested by the analyses presented in Figures 2 and 3. To generate an embedding comparison that provides direct evidence for the absence of distinct gaits, it would be necessary to have a model for the structure of gait transitions. However, there does not exist *a priori* a clear null model for intermediate coordination patterns, nor does our experimental data suggest such a model. In the absence of such a null model, Figure 4—figure supplement 1 presents the case in which transitions do not occur within the embedded segments of coordination patterns. We regret that we did not make this feature of the analysis clear in the original manuscript; in particular, the order in which the synthetic and empirical data embeddings were presented framed this analysis as a stronger hypothesis test than we believe it is. We have therefore revised the ordering of this section of the manuscript, and have added prose to clarify the intent of this analysis.

Additionally, we would like to clarify that we do not use the binary swing/stance variables to estimate limb phases; rather, we estimate limb phases directly from the raw limb positional time series using the discrete-time analytic signal method. As described in our response to Essential Revision 3, our investigation of the noise sensitivity of this method using synthetic data shows that measurement error leads to broader peaks in phase distributions but does not change the presence or locations of those peaks.

4) A common feature in the gait transition literature is hysteresis – the transition occurs at a different speed going from slow-to fast than fast-to-slow. All of the analyses here fundamentally assume that gait is a single-valued function of speed, however, which may cause for some misinterpretations of the data, since all spread of the distribution would just be assigned to a single mean value. Either adapting the analysis to include this potential effect (or explaining why it is not relevant here) is a necessary addition to the manuscript.

Our primary response is in the comments to Essential Revision 5. Based on our additional analysis, we have not found evidence for hysteresis in our dataset.

Reviewer #2:In this paper, DeAngelis and colleagues investigate the temporal organization of the fly walking gait. They use a simple tracking algorithm to measure the body orientation and distal leg position of flies walking in a flat, featureless environment. They confirm previous observations that stance duration changes as a function of walking speed. They then extracted swing and stance phases from each leg and used these patterns to classify different walking gaits. They find that flies typically have three feet in stance (tripod gait), but they observe that flies occasionally (and transiently) have 4 or 5 feet in stance, particularly at low walking speeds. Contralateral limbs reliably swing in antiphase, while ipsilateral limb coordination vary a bit more. A low-dimensional embedding visualization of limb coordinates resulted in cyclic (global oscillator phase) and one linear coordinate (stepping frequency). Overall, this suite of analyses suggest that there is no clear differentiation of walking behavior into distinct gaits (e.g., tripod vs. tetrapod). The authors then construct a model that can predict the continuum of walking behavior by tuning of a single parameter. Finally, they find that perturbing walking behavior through activation of the Moonwalker Descending Neuron or visual stimulation alters stance duration, and thus induces a shift along the length of the walking manifold.Overall, the paper is clearly organized and well-written. Although the subject of the fly walking gait is already well-trodden, this is certainly the most exhaustive and definitive treatment to date. I think it is a very strong paper and have only a handful of suggestions to improve it prior to publication.1) My biggest point of confusion was the coupled phase oscillator model. The paper would be improved if this model were motivated and presented more clearly. The equations are not intuitive at first glance (e.g., the 1/sin terms, the discontinuous indicator functions); is there an easy interpretation of them? And does the specific form of the model matter? How is it related to the other coupled oscillator models in the literature? I would have liked to see a sensitivity analysis that provides some intuition for how the model works and which parameters matter. For example, τ_stance_ is the relevant parameter that the authors tune to get different inter-limb coordination patterns, but this is not clearly explained (especially in the main text). Finally, how does this model contribute to our understanding or intuition of the control strategy used by the fly to coordinate walking? The authors should include a deeper discussion of the motivation for and interpretation of this model.

Thank you for this comment. We have addressed this comment in detail in the reply to Essential Revision 4 above.

Reviewer #3:This paper is notable for collecting a very comprehensive dataset of kinematic gait patterns in *Drosophila* over a wide range of spontaneous and evoked speed changes. While the optogenetic and visually induced changes are nice as supporting evidence, the core of the paper comes from a detailed and thorough gait analysis. As such the paper has some very good insights when connected to modern dimensionality reduction and some simple models of oscillations, but needs to distinguish itself carefully from the long history of papers that also do gait analysis especially in *Drosophila*.1) The structure of the argument is that animals in general are thought to have distinct gaits. The authors cite a number of studies on larger legged animals for this, but in those cases walking is an out of phase oscillation of gravitational potential energy and kinetic energy (the inverted pendulum model) and running is in-phase oscillations (the spring-loaded inverted pendulum). The transition in gaits in many animals from insects to horses corresponds to the transition between these underlying dynamics If this transition is unlikely (although possible?) for *Drosophila* given their size, it is not clear why we would expect distinct gaits even if larger animals have them.More critical to the conclusions drawn about *Drosophila*, previous studies (Wosnitza, et al., 2012, and Mendes, et al., 2013) My reading of these previous studies is that they show that limb coordination patterns in *Drosophila* are highly variable and that while there were changes with speed that the overall shifts are more continuous than observed in many other animals. These are cited in the paper but the relationship of their conclusions to this paper are only discussed superficially. For example Wosnitza, et al., 2012 conclude that their findings "imply that *Drosophila*'s walking behavior is more flexible than previously thought (Strauss and Heisenberg, 1990): there are no clearly separable gaits and, more specifically, the neural controller producing inter-leg coordination is not restricted to a fixed tripod pattern." This is very close to the initial conclusions of this paper.So while some data from those studies do fit tetrapod and tripod gaits (as do some in this study), the conclusion that *Drosophila* use a variety kinematic patterns that encompass the canonical gaits is already somewhat established. Therefore the hypothesis of distinct gaits in this paper seems like a bit of a strawman. The authors do make some nice analyses of these data and show the point perhaps more convincingly than in previous work. However, prior to the manifold section, the initial presentation of the data did not leave me thinking that I had learned something new. I recognize that the authors may not agree with interpretations in previous works, but the onus is on the authors to clearly articulate what cannot be concluded from the previous work that their work in turn justifies.More importantly this whole argument of distinct gaits takes away from what for me was the real impact of the paper. The central advance of the paper is the finding that kinematics patterns across a large range of speeds can collapse down onto a manifold with a single parameter governing speed variation. The authors are still making an advance here, but the motivation of the introduction and the conclusion of the first sentence of the Discussion highlight the continuum vs. discrete nature of *Drosophila* gaits. The emphasis should be about the manifold.

Our primary response to this comment is with Essential Revision 3. We have followed the suggestions of the reviewer and made the manifold structure the primary emphasis of the paper.

2) My concern about the manifold discussion is how this approach is different from the many other dimensionality reductions that have been done on gait. What really sets it apart from the t-SNE analyses of behavior that show all walking lying in a low dimensional cluster (Berman, et al., 2014) and the reduced phase and oscillator models of stick insects and cockroaches (e.g. Couzin-Fuchs, et al., 2015). I think there is novelty here but it must be articulated more in context.

Our main response to this is included with Essential Revision 1. We have updated the prose to expand our discussion of the relative benefits of UMAP over previous dimensionality reduction techniques. In particular, because t-SNE is designed to emphasize local rather than global structure, it does not immediately capture the periodic dynamics of legged locomotion, which UMAP highlights (Becht et al., 2019; McInnes et al., 2018). Furthermore, UMAP has been shown to produce embeddings that are more reproduceable across data samples and subsample sizes than those obtained using t-SNE (Becht et al., 2019).

To compare the results of applying t-SNE with those obtained using UMAP, we used the fast interpolation-based algorithm for t-SNE introduced in (Linderman et al., 2017) to generate embeddings of our limb kinematic data. Following standard practices, we used the standard target dimensionality for t-SNE (two) and a perplexity of 100, and reduced the dimensionality of the input data to 50 using PCA prior to applying t-SNE (Becht et al., 2019; Berman et al., 2014; Linderman et al., 2017; Van Der Maaten, 2014). As shown in Author response image 4-SNE does capture some of the frequency structure of the manifold produced by UMAP, but, unlike UMAP, it produces an embedding that appears segmented.

We wanted to better understand whether this segmentation arose from features of the data or features of the algorithm. We therefore applied the identical analysis to data generated by our model. By construction, the high-dimensional data manifold produced by the model is continuous. However, the t-SNE embedding of the model data is also segmented, suggesting that the structure observed in the embedding of experimental data could result from the analysis itself rather than the underlying structure of the data (Author response image 4). This result makes sense given that t-SNE is explicitly designed to emphasize local similarities in the high-dimensional data at the expense of large-scale structure (Van Der Maaten, 2014). This is in some ways opposite to the purpose of UMAP, which is designed to preserve both local and large scale structure in the data. Therefore, unlike UMAP, the application of t-SNE to our limb kinematic data does not produce results that are immediately interpretable. Since the goal of our dimensionality reduction analysis is to visualize the global structure of our data, the interpretability of the embedding is key.

**Author response image 4. respfig4:** t-SNE embedding of limb position data, colored by forward walking speed. (**A**) shows experimental limb positional data, while (**B**) shows data generated using the continuum model presented in Figure 5 of the manuscript.